# The interindividual variability of multimodal brain connectivity maintains spatial heterogeneity and relates to tissue microstructure

Esin Karahan[1], Luke Tait [2], Ruoguang Si[1], Ayşegül Özkan[1], Maciek J. Szul [3,4], Kim S. Graham[5], Andrew D. Lawrence [1] & Jiaxiang Zhang [1,6✉]

Humans differ from each other in a wide range of biometrics, but to what extent brain connectivity varies between individuals remains largely unknown. By combining diffusion-weighted imaging (DWI) and magnetoencephalography (MEG), this study characterizes the inter-subject variability (ISV) of multimodal brain connectivity. Structural connectivity is characterized by higher ISV in association cortices including the core multiple-demand network and lower ISV in the sensorimotor cortex. MEG ISV exhibits frequency-dependent signatures, and the extent of MEG ISV is consistent with that of structural connectivity ISV in selective macroscopic cortical clusters. Across the cortex, the ISVs of structural connectivity and beta-band MEG functional connectivity are negatively associated with cortical myelin content indexed by the quantitative T1 relaxation rate measured by high-resolution 7 T MRI. Furthermore, MEG ISV from alpha to gamma bands relates to the hindrance and restriction of the white-matter tissue estimated by DWI microstructural models. Our findings depict the inter-relationship between the ISV of brain connectivity from multiple modalities, and highlight the role of tissue microstructure underpinning the ISV.

[1] Cardiff University Brain Research Imaging Centre, School of Psychology, Cardiff University, Cardiff, United Kingdom. [2] Centre for Systems Modelling and Quantitative Biomedicine, University of Birmingham, Birmingham, United Kingdom. [3] Institut des Sciences Cognitives Marc Jeannerod, CNRS UMR 5229 Bron, France. [4] Université Claude Bernard Lyon I, Lyon, France. [5] Department of Psychology, University of Edinburgh, Edinburgh, United Kingdom. [6] Department of Computer Science, Swansea University, Swansea, United Kingdom. ✉email: zhangj73@cardiff.ac.uk

Humans vary in their biometrics such as DNA, fingerprints, and iris patterns, which allow them to be used for identifying an individual. Similarly, connectome, the wiring patterns between brain regions, exhibit substantial variability among individuals at anatomical[1], structural[2], functional[3] and neurophysiological levels[4]. Within an individual, the brain connectome further changes throughout the life span[5] and undergoes profound modifications in many neurological disorders[6]. Therefore, understanding the inter-subject variability (ISV) of connectivity is necessary for establishing a link between brain, cognition and typical or atypical lifespan development.

For fMRI resting-state functional connectivity, there is higher ISV in the frontoparietal network than in the rest of the cortex[2,7]. For DWI-based structural connectivity, the optic radiation has a higher ISV than the corticospinal tract[8]. The spatial heterogeneity of the ISV has been associated with the regional difference in cortical surface expansion[9] and cortical folding[10] during development. However, research on the ISV of brain connectivity raised two open questions.

First, although the ISV of fMRI functional connectivity is well studied[2,7], there is a lack of understanding of the ISV of brain connectivity in other imaging modalities. DWI- and MEG-based connectivity quantify brain networks in different spatiotemporal resolutions from BOLD fMRI, and they reflect distinct neurobiological underpinnings. MEG functional connectivity captures the frequency-dependent oscillatory coupling of macroscopic neural activity[4], and streamline tractography on DWI data estimates the strength of white matter pathways[11]. It is unknown whether the ISV of DWI- and MEG-based connectivity exhibit similar spatial heterogeneity as that of fMRI.

Second, although the ISV of brain connectivity describes the individual difference in connectome, less is known about whether grey and white matter microstructure may give rise to such variability in connectivity. Previous research implies a close relationship between tissue microstructure and connectivity. For example, in grey matter, cortical myelin content delineates the border of brain areas[12,13] and can predict brain connectivity[14]. In white matter, microstructural properties of fibre tracts predict cross-subject variance in the interhemispheric functional connectivity between homotopic regions[15]. More recent evidence is from non-human primates, which showed that the ISV of fMRI resting-state connectivity in awake macaque monkeys relates to the T1w/T2w ratio map, an in vivo contrast sensitive to intracortical myelin[16]. However, there is currently no systematic evidence of the relationship between the ISV of human brain connectivity and tissue microstructure.

Here, we addressed these questions by systematically characterizing the ISV of whole-brain connectivity using a multimodal imaging dataset including DWI and MEG. We took a unique approach to facilitate comparisons between imaging modalities while at the same time accounting for the spatial variations of the signal-to-noise ratio in MEG[17]. We generated DWI-based connectomes based on an upsampled version of the HCP-MMP atlas[13] (Fig. 1a), and source-localized MEG-based connectome based on a down-sampled version of the HCP-MMP atlas[17], optimized for MEG source-level signal-to-noise ratio (Supplementary Fig. 1). This approach allowed us to evaluate cross-modal alignment at the macroscopic cortical cluster level (Supplementary Fig. 2) while maintaining the sensitivity of connectivity measure in each modality. Across the cortex, we then related the ISV of multimodal connectome (Fig. 1b) to (1) cortical myelin content, indexed by the T1 relaxation rate measured by high-resolution 7 T MRI, and (2) the hindrance and tissue complexity of fibre tracts, indexed by white matter compartment models and the diffusion tensor model. We further assessed the reliability of the structural connectivity's ISV by replicating our results on an independent open-access dataset.

We observed that both structural and MEG functional connectivity exhibit non-homogeneous ISV across the cortex. The extent of ISV is maintained between imaging modalities in selective cortical clusters. Both grey-matter and white-matter microstructure are associated with the ISV of connectivity. Our results highlighted the interlink between cohort-level connectivity variability and tissue properties.

## Results

**Inter-subject variability of multimodal connectome varies across the cortex.** We calculated the multimodal connectomes of individual participants from DWI tractography (Fig. 2a, b) and from resting MEG in correlations of oscillatory power envelops (Fig. 2c–f). The DWI-based structural connectivity matrices were constructed from an upsampled HCP-MMP atlas[13], which contains 664 regions of interest (ROIs) with similar sizes to minimise the confound of regional size in estimating connectivity variability[18]. The functional connectivity matrices of MEG data were constructed from a downsampled HCP-MMP atlas with 230 ROIs, which is optimized for MEG source-space signal-to-noise ratio[17]. To facilitate comparisons between modalities, both the upsampled and downsampled atlases maintained the macroscopic structure of 22 bilateral cortical clusters in the original HCP-MMP atlas[13] (Supplementary Fig. 2).

For each brain region, we quantified the inter-subject variability (ISV) of its connectivity profile under each imaging modality. The ISV of a region is defined as the mean cosine distance between the region's connectivity profiles from all pairs of participants (Fig. 1b). We then quantified the ISV of each of the 22 cortical clusters in the HCP-MMP atlas, by averaging the ISVs of all regions within the cluster.

Among cortical clusters, the ISV of structural connectome (SC-ISV) was high in the frontal, parietal, and cingulate cortices. The SC-ISV was low in the unimodal sensorimotor cortices, including V1 and early visual cortices, early and associate auditory cortices, as well as somatosensory and motor cortices. We further analyzed the SC-ISV of an independent dataset from the Cam-CAN repository[19] (Fig. 2b). The SC-ISV maps derived from our data and the Cam-CAN dataset were consistent across cortical ROIs ($R = 0.76$, 95% CI = [0.72, 0.79], $p = 0.0002$ SA-corrected, $p = 2.51 \times 10^{-124}$ uncorrected, Spearman's correlation, see also Supplementary Fig. 3 for the correction of spatial autocorrelation), suggesting that the ISV of structural connectivity is robust and generalizable between cohorts.

We calculated the MEG-based ISV (MEG-ISV) from the MEG functional connectivity matrices from two recording sessions to correct for inter-session variability. MEG-ISV was characterized by high variability in frontal clusters in theta, alpha, and beta bands (Fig. 2c–e). Theta and alpha band MEG-ISV were low in visual clusters, whereas beta-band MEG-ISV was low in somatosensory and motor clusters. The MEG-ISV of gamma band connectivity (Fig. 2f) differed from that of the other frequency bands, with high variability in the parietal cortex and low variability in the medial and lateral temporal cortices. Therefore, we observed non-uniformly distributed ISVs across the cortical surface in all modalities.

**Cross-modal correspondence of inter-subject connectivity variability is frequency dependent in MEG.** Could the extent of the ISV be consistent between imaging modalities? To test this hypothesis, for each cortical cluster, we calculated the Spearman rank correlation between the distances in DWI and MEG connectivity profiles across all pairs of participants. This allowed for cross-modal comparisons at the cluster level, with cortical parcellation for DWI and MEG data at different spatial resolutions.

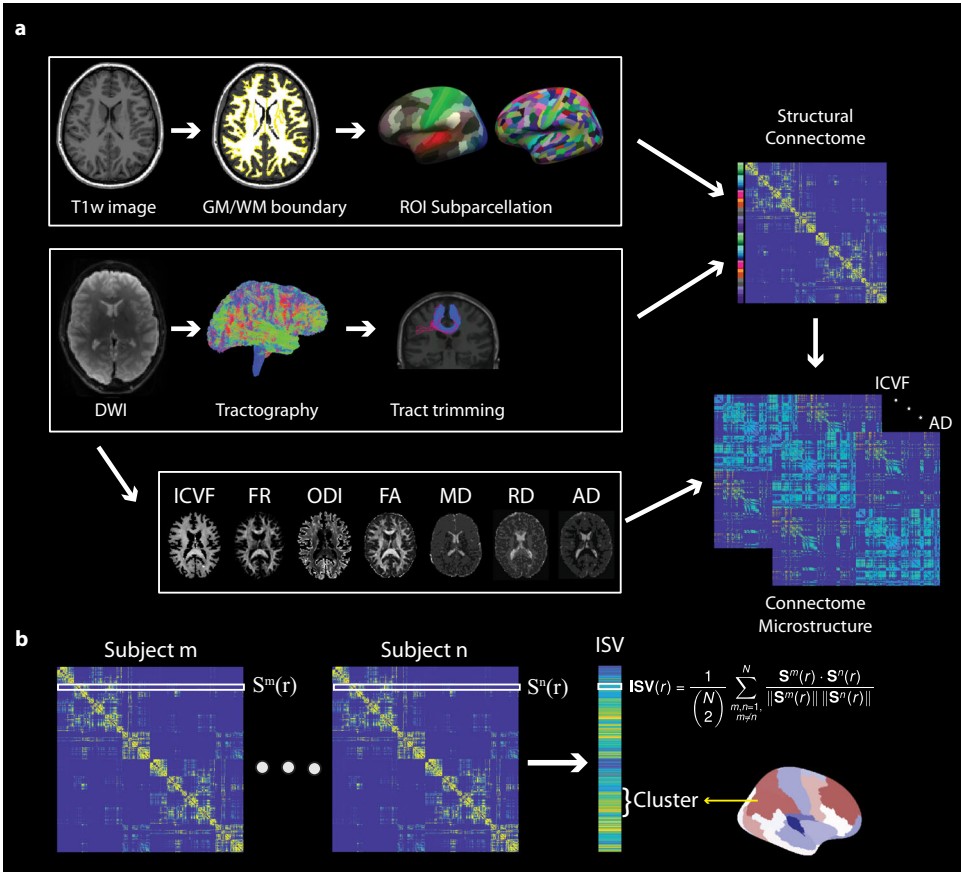

**Fig. 1 Analysis pipeline of the structural connectome and inter-subject variability (ISV). a** For each participant, after pre-processing of T1w images, the grey matter/white matter (GM/WM) boundary was extracted. HCP-MMP upsampled atlas was registered to the individual space and sampled into volume. After pre-processing of DWI data, whole-brain probabilistic tractography was performed. Streamlines from tractography were trimmed using a clustering-based method. Finally, a structural connectivity matrix was obtained. Microstructural measures from DTI, CHARMED, and NODDI models were calculated and sampled along streamlines to quantify the microstructural properties of structural connectivity. **b** For each imaging modality, the ISV per ROI was defined as the mean cosine distance between individuals' connectivity matrices across all pairs of subjects. We then took the average ISV measures from all ROIs within an HCP-MMP cortical cluster to facilitate comparisons between imaging modalities.

In all frequencies, MEG-ISVs had significant correspondences to the SC-ISV in the lateral temporal cortex (Fig. 3a–d, $p < 0.05$, FWE corrected). There were also wider correlations between the SC-ISV and beta band MEG-ISV in the early visual, early auditory, parietal, and prefrontal clusters (Fig. 3c). In sum, the ISV of brain connectivity is consistent between modalities in selective cortical clusters, and some cross-modal correspondences are frequency dependent in MEG-ISV.

**Highly myelinated cortical regions have low inter-subject connectivity variability.** Grey-matter myelination varies substantially across the cortex and is critical in mediating synaptic plasticity[20], connectivity[21] and behaviour[22]. Cortical myelin content provides critical information for the parcellation of the cerebral cortex, because regions with similar cyto- and myeloarchitecture tend to connect with each other both functionally and structurally[11,12,14].

To test whether cortical myelin content relates to the ISV of connectivity, we acquired submillimetre (650 μm) whole-brain maps of T1 relaxation rate (R1) at 7 T from an independent age- and gender-matched cohort (see Cohort 2 in Method). R1 is a quantitative MR measure sensitive to the myelin content in the grey matter, validated by postmortem histological studies[23,24].

R1-derived cortical myelin maps showed a clear distinction between association and sensorimotor cortices. The myelin content was high (i.e., large R1 values) in the primary visual, motor and early auditory clusters, and it was low (i.e., small R1 values) in the medial and lateral prefrontal clusters (Fig. 4a).

Vertex-wise R1 maps were parcellated according to the upsampled and downsampled HCP-MMP atlas to match the ISV maps. Across the cortex, regions with higher cortical myelin content, as indexed by the R1 value, are associated with lower SC-ISV of Cohort 1 data ($R = -0.25$, 95% CI = [−0.32, −0.17], $p = 0.02$ SA-corrected, $p = 2.56 \times 10^{-10}$ uncorrected). The negative association between SC-ISV and R1 value is replicated in Cam-CAN dataset ($R = -0.45$, 95% CI = [−0.52, −0.38], $p = 0.0002$ SA-corrected, $p = 7.49 \times 10^{-32}$ uncorrected) (Fig. 4b and Supplementary Fig. 3).

Beta-band MEG-ISV negatively correlated with the R1 value across the cortex ($R = -0.35$, 95% CI = [−0.47, −0.23], $p = 0.04$ SA-corrected, $p = 1.27 \times 10^{-7}$ uncorrected) (Fig. 4c and Supplementary Fig. 4a). MEG-ISV at theta and alpha bands showed a similar trend, but the results were not significant after correction for spatial autocorrelation (theta MEG-ISV: $R = -0.22$, 95% CI = [−0.34, −0.09], $p = 0.32$ SA-corrected, $p = 0.001$ uncorrected; alpha MEG-ISV: $R = -0.23$, 95% CI = [−0.35, −0.10], $p = 0.45$ SA-corrected, $p = 0.0007$ uncorrected). There is no relationship between gamma-band MEG-ISV and R1 values ($R = 0.13$, 95% CI = [−0.005, 0.26], $p = 0.49$ SA-corrected, $p = 0.06$ uncorrected). That is, for brain regions with lower cortical myelin content, their structural connectivity and beta-

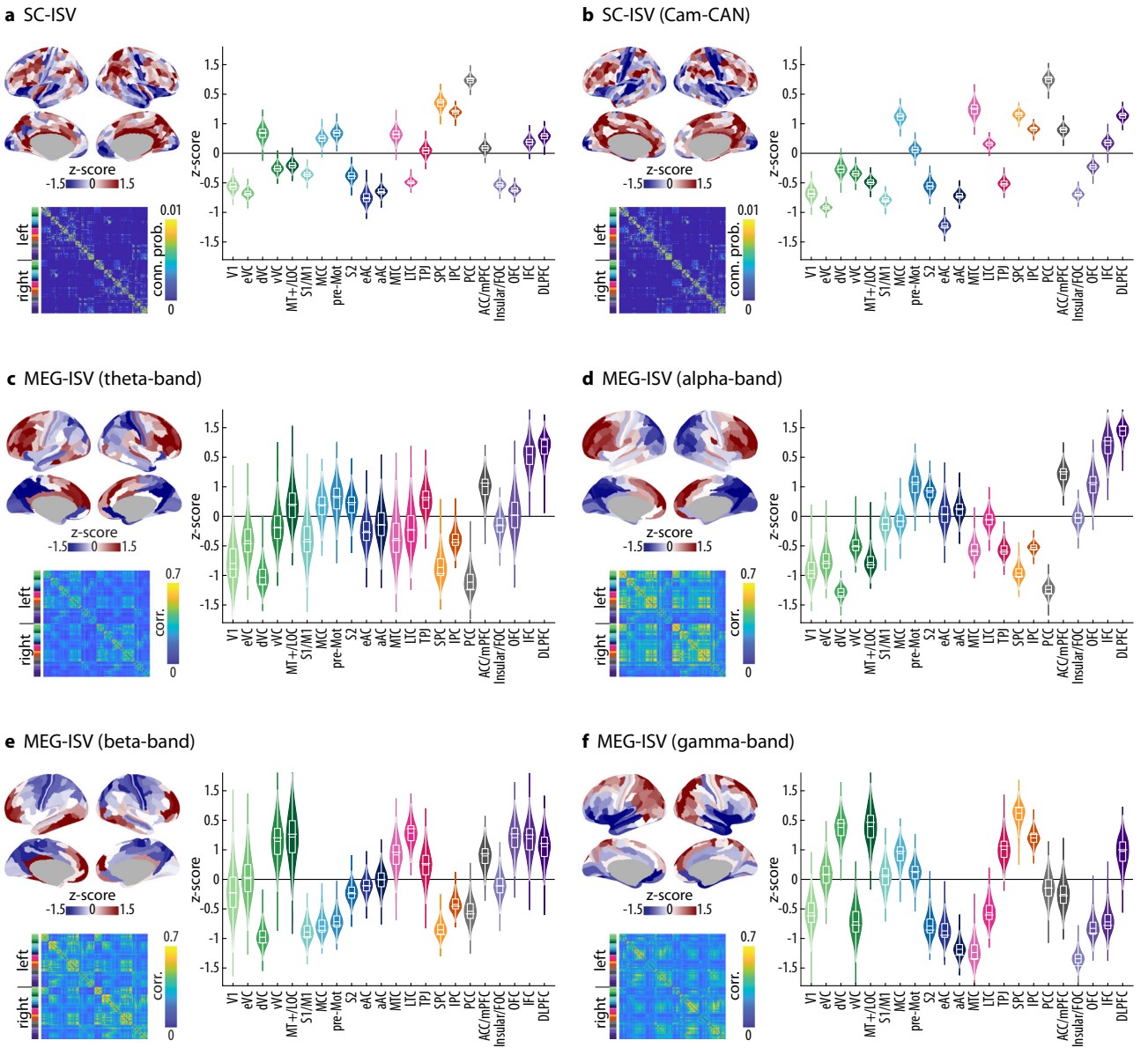

**Fig. 2 Inter-subject variability (ISV) of brain connectivity. a–b** The ISV of structural connectivity from Cohort 1 (**a**) and Cam-CAN (**b**). **c–f** The ISV of MEG functional connectivity in each frequency band. In each panel, the brain maps (upper left) display the z-score of each ROI's ISV on the cortical surface. SC-ISV is illustrated on an upsampled HCP-MMP atlas of 664 ROIs. MEG-ISV is illustrated on a downsampled HCP-MMP atlas[17] of 230 ROIs. Violin plots (right) show the distribution of the ISV of each cortical cluster, which was estimated from all ROIs within each cluster through a bootstrapping procedure (5000 bootstrap samples). In each violin plot, the horizontal bar denotes the observed ISV at the cluster level. The nested box plot shows the median and the interquartile range (IQR) of the bootstrap distribution. The whiskers are 1.5 times the IQR. The connectivity matrix (lower left) denotes the group average of connectomes. DWI-based structural connectivity is measured by the connection probability from tractography. MEG-based function connectivity is measured by amplitude envelope correlation within each frequency band. The columns and rows of the connectivity matrices are ordered as follows: (1) left-hemisphere ROIs are before right-hemisphere ROIs, (2) ROIs belonging to each cortical cluster are grouped together, and (3) the 22 clusters follow the same order as in the violin plot. Different colours next to the connectivity matrix and in the violin plot represent the 22 different cortical clusters. Supplementary Fig. 2 illustrates the anatomical locations of the 22 clusters and the full name of each cluster's acronym.

band MEG functional connectivity are more variable between participants.

**White matter microstructure relates to the inter-subject variability of MEG functional connectivity**. The covariation of tissue microstructure and functional connectivity[15] suggests a possible relation between white-matter microstructural metrics and inter-subject variability. For each pair of regions in the structural connectivity matrix, we fitted two tissue-compartment

models (NODDI and CHARMED) and the conventional DTI model to the DWI data, yielding 7 microstructural metrics along each pair's white matter tracts: intracellular volume fraction (ICVF) and orientation dispersion index (ODI) from the NODDI model; restricted water fraction from the CHARMED model (FR); as well as FA, MD, AD and RD from the DTI model. Because these microstructural metrics are not independent of each other[25], we performed PCA on the microstructural metrics to reduce data dimensionality and obtain biologically relevant principal components[26].

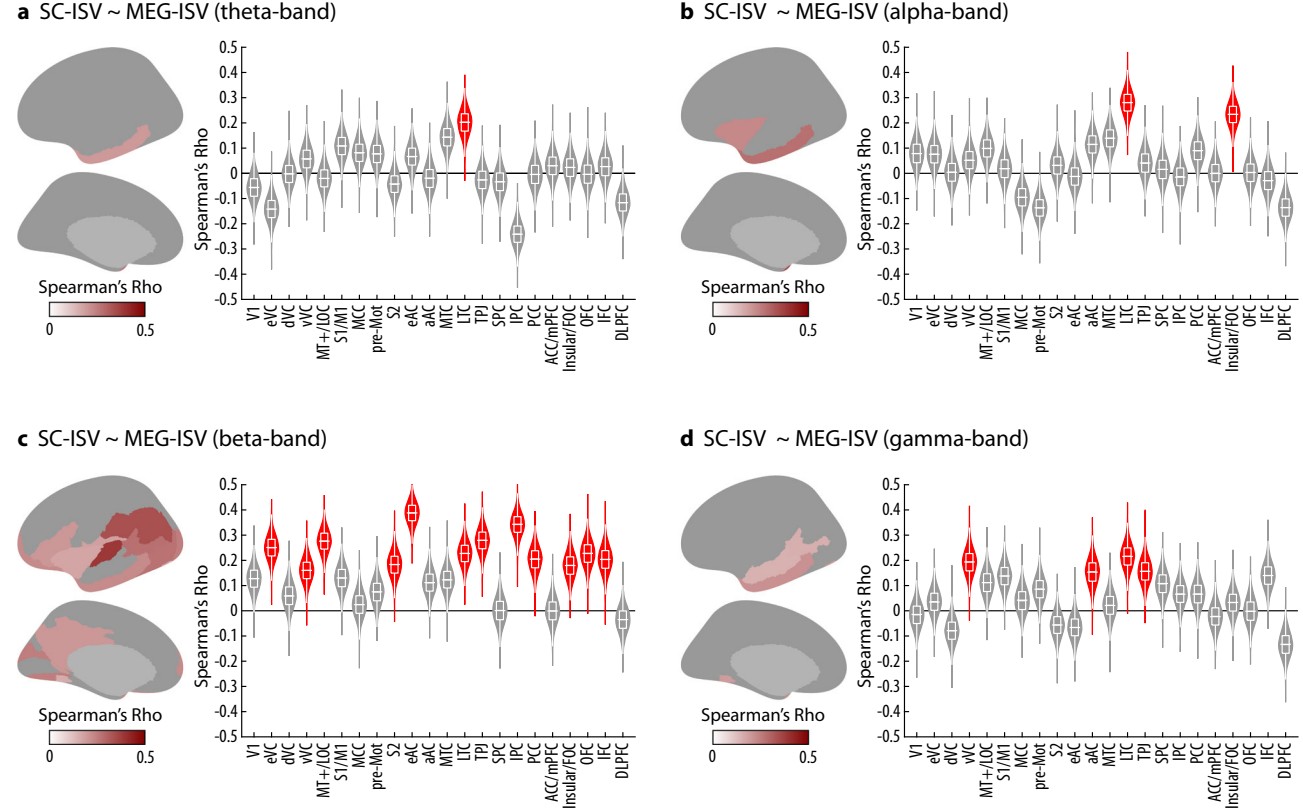

**Fig. 3 Cross-modal correspondence between SC-ISV and MEG-ISV in each MEG frequency band. a** Theta-band MEG-ISV. **b** Alpha-band MEG-ISV.
**c** Beta-band MEG-ISV. **d** Gamma-band MEG-ISV. In each panel, the violin plots show the bootstrap distribution and observed statistics (horizontal bars) of
Spearman's correlation coefficients between SC-ISV and MEG-ISV in individual clusters. Each nested box plot shows the median and the interquartile range
(IQR) of the bootstrap distribution. The whiskers are 1.5 times the IQR. Red violin plots denote clusters with significant cross-modal correspondence
($p < 0.05$ FWE corrected, 10,000 permutations). The brain map shows the anatomical locations of the significant clusters and their correlation coefficients
on the cortical surface.

Across participants and white-matter regional connectivity, two principal components explained >90% of the variance (Fig. 5a, b and Supplementary Fig. 5). The first principal component (PC1) described the hindrance and restriction of the white-matter tissue (explained variance, 55.00%), with positive loadings of FR (loading coefficient 0.42), ICVF (0.36), FA (0.48) and AD (0.30), as well as negative loadings of ODI (−0.43), RD (−0.43) and MD (−0.04). The second principal component (PC2) relates to tissue complexity (explained variance, 35.77%), with positive contributions from MD (0.61), AD (0.50), FA (0.19) and RD (0.26), as well as negative loadings of ICVF (−0.34), ODI (−0.31) and FR (−0.26).

We summarized the tissue microstructure of each region's structural connectivity by averaging PC1 or PC2 values across all tractography streamlines originated from the region (Fig. 5a). Across the cortex, the value of PC1 was high in structural connections originating from the pre-motor, parietal, and posterior cingulate cortices. In contrast, the value of PC2 was high in structural connections originating from the visual and temporal cortices, medial temporal regions and the frontopolar cortex.

We re-parcellated the PC1 and PC2 maps using the downsampled HCP-MMP atlas to match the resolution of MEG-ISV. Across the cortex, PC1 values were negatively associated with MEG-ISV in the alpha ($R = −0.62$, 95% CI = [−0.70, −0.52], $p = 0.045$ SA-corrected, $p = 2.51 \times 10^{-25}$ uncorrected) and beta bands ($R = −0.42$, 95% CI = [−0.53, −0.30], $p = 0.03$ SA-corrected, $p = 3.27 \times 10^{-11}$ uncorrected), and positively correlated with gamma-band MEG-ISV ($R = 0.55$, 95% CI = [0.45, 0.64], $p = 0.004$ SA-corrected,

$p = 7.83 \times 10^{-20}$ uncorrected) (Fig. 5c and Supplementary Fig. 4b). There was no significant relationship between PC1 values and theta-band MEG-ISV ($R = −0.34$, 95% CI = [−0.48, −0.21], $p = 0.17$ SA-corrected, $p = 9.02 \times 10^{-8}$ uncorrected). After correcting for the spatial autocorrelation in brain maps, there was no significant relationship between PC2 values and MEG-ISV (theta: $R = −0.31$, 95% CI = [−0.41, −0.19], $p = 0.22$ SA-corrected, $p = 2.19 \times 10^{-6}$ uncorrected; alpha: $R = −0.28$, 95% CI = [−0.38, −0.18], $p = 0.44$ SA-corrected, $p = 1.62 \times 10^{-5}$ uncorrected; beta: $R = 0.38$, 95% CI = [0.28, 0.48], $p = 0.06$ SA-corrected, $p = 1.65 \times 10^{-9}$ uncorrected; gamma: $R = −0.32$, 95% CI = [−0.44, −0.20], $p = 0.13$ SA-corrected, $p = 5.34 \times 10^{-7}$ uncorrected) (Fig. 5c and Supplementary Fig. 4c).

**The core Multiple Demand (MD) network has high inter-subject variability of structural connectivity.** The analysis above showed how the ISV of MEG functional connectivity relates to DWI-based microstructural properties. Does the ISV of structural connectivity have functional significance? To address this question, we considered the SC-ISV of the MD network. The MD network is a domain-general system that is consistently activated by many different types of cognitive demands, such as memory, problem-solving and attention[27]. Figure 6a showed the most recent definition of the MD network on the HCP-MMP atlas, including core MD regions that are mostly functionally inter-connected, surrounded by a penumbra of peripheral regions[28]. We compared the mean SC-ISV of the core MD regions and the penumbra with the rest of the cortex (non-MD regions). In both cohorts, the core MD regions had higher SC-ISV than non-MD regions (Cohort 1: $p = 0.0006$; Cam-CAN: $p = 0.0006$, Bonferroni

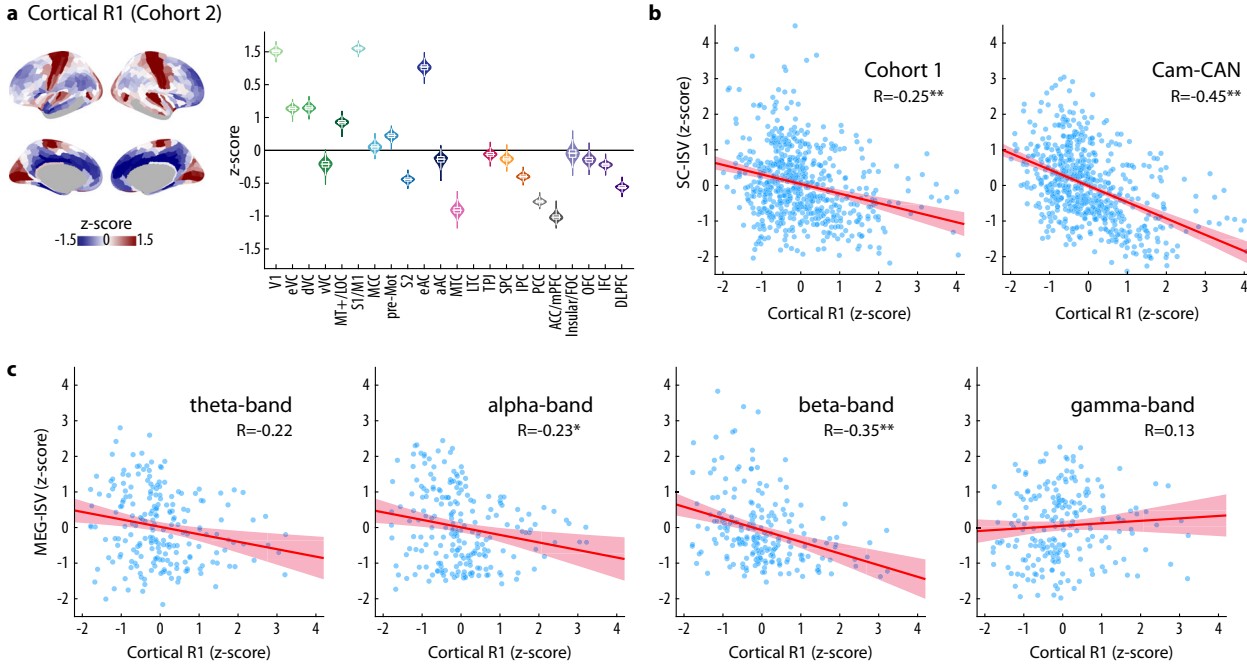

**Fig. 4 Inter-subject variability of connectivity relates to cortical T1 relaxation rate (R1). a** Left: The group average of R1 values from Cohort 2 is converted to z-scores and plotted on the cortical surface. Right: The z-score of the R1 value of each cluster. The grouping of ROIs and the definition of violin plots are the same as in Fig. 2. **b** Correlations between SC-ISV (left: Cohort 1; right: Cam-CAN) and R1 values across ROIs. **c** Correlations between MEG-ISV and R1 values across ROIs. In panels **b**, **c**, the regression lines (solid) and the 95% confidence intervals (shaded area) are shown. *R* represents the Spearman rank correlation coefficient. Asterisks denote statistical significance (*$p < 0.001$ uncorrected for spatial autocorrelation, **$p < 0.05$ SA-corrected).

corrected, permutation tests; Fig. 6b). In Cohort 1, core MD regions had higher SC-ISV than the penumbra ($p = 0.004$). In the Cam-CAN dataset, the penumbra had higher SC-ISV than non-MD regions ($p = 0.003$).

## Discussion

We systematically quantified the inter-subject variability (ISV) of whole-brain structural and functional connectivity, examined its cross-modal consistency, and related its spatial characteristics to grey and white matter microstructural properties. DWI and MEG connectivity variability is consistent in selective cortical clusters, as supported by significant cross-modal ISV alignments. The spatial heterogeneity of both structural and beta-band MEG connectivity ISVs is associated with that of the cortical myelin content. Alpha, beta and gamma band MEG-ISVs further relate to white matter microstructure. Our findings extended the current understanding of brain connectivity variability in multiple modalities and suggested the important roles of tissue microstructure in shaping connectivity variability.

Brain variability exists in different forms[29]. For example, one can quantify anatomical variability as the amount of deformation between individual brains and a group template. In both humans and non-human primates, the anatomical variability in visual and frontal areas was higher than in other brain regions[30]. FMRI localisation and cytoarchitectonic classification studies showed that visual[31] and motor cortices[32] have high morphometric variability. These brain variability measures differ from the results of the current study, which focused on the ISV of structural and functional connectivity.

The ISV of DWI- and MEG-based connectivity exhibited spatial heterogeneity across the cortex. The SC-ISV in hetero-modal association cortices (frontal, parietal, and cingulate areas) are higher than those in unimodal sensorimotor cortices. We further replicated this result in an independent dataset (Cam-

CAN), and a similar pattern has been reported elsewhere using different acquisition parameters, atlases and pre-processing methods[2,18,33].

The spatial distribution of SC-ISV carries functional significance: the SC-ISV is high in the core MD regions. Core MD regions in the frontoparietal cortex are essential for cognitive flexibility and integration, and they have been linked to individual difference in memory and fluid intelligence[34]. Here, we extended previous findings that core MD regions have strong interconnectivity[28], by showing core MD regions to have greater connectivity variability than the rest of the cortex.

The anatomical signature of SC-ISV's unimodal-heteromodal distinction resembles the principal gradient of cortical hierarchy that is related to synaptic physiology and cytoarchitecture[14]. Such a spatial pattern is also reported in human brain expansion during evolution and postnatal development[10,35]. Multimodal association cortices are evolutionary modern and present a higher surface expansion in humans with respect to non-human primates[36], specifically in the frontoparietal network[9]. Interestingly, those large expanding cortical regions are less developed at term gestation and mature both structurally and functionally throughout development[37], potentially allowing them to be influenced by environmental factors[37,38]. Therefore, the high SC-ISV in association cortices could be due to cortical surface expansion accompanied with local cellular events, with late maturation in the white matter[39] and synaptic density[37]. Evidence from developmental neuroscience support this proposition. Regional differences in white matter maturation in later childhood development lead to variability in structural connectome[40]. Furthermore, individual cognitive abilities relate to the surface size of brain regions exhibiting high expansion during evolution and development[35].

The SC-ISV of some regions did not follow the principal axis of unimodal-association cortices. Most notably, the orbital and polar

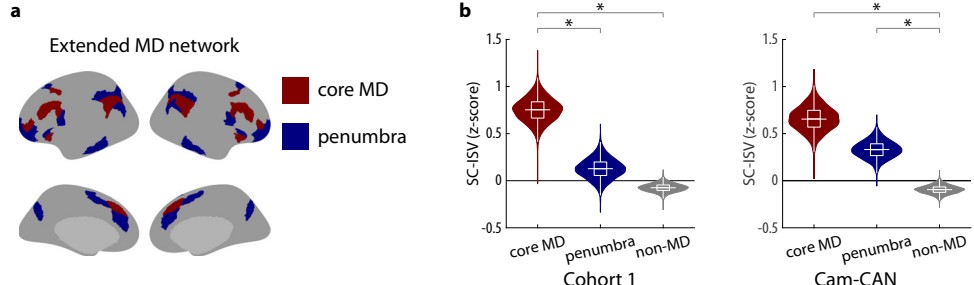

**Fig. 5 Tissue microstructure of structural connectome and its relations to MEG-ISV. a** The first (PC1) and second (PC2) principal components of microstructural metrics averaged across participants. The PC1 and PC2 values are converted to z-score and rendered on the cortical surface. **b** PCA results with 7 microstructural metrics. Left: the loading of the first two principal components on each microstructural metric. Right: Pearson correlation coefficients between microstructural metrics across the connectivity matrix. ICVF, intracellular volume fraction; ODI, orientation dispersion index; FR, restricted water fraction; FA, fractional anisotropy; MD, mean diffusivity; AD, axial diffusivity; RD, radial diffusivity. **c** Scatter plots show the MEG-ISV as a function of the PC1 and PC2 values in each cortical region. The regression line (solid) and the 95% confidence intervals (shaded area) are shown. *R* represents the Spearman rank correlation coefficient. Asterisks denote statistical significance (*$p < 0.001$ uncorrected for spatial autocorrelation, **$p < 0.05$ SA-corrected).

**Fig. 6 SC-ISV in the multiple demand (MD) network. a** The extended MD network includes the core MD regions and the penumbra. All MD regions were defined in the HCP-MMP atlas[28]. **b** The SC-ISV of the core MD regions, the penumbra and non-MD regions from Cohort 1 (left) and Cam-CAN (right) datasets. Violin plots show the full bootstrap distribution and observed values (horizontal bars) of the SC-ISV. Each nested box plot shows the median and the interquartile range (IQR) of the bootstrap distribution. The whiskers are 1.5 times the IQR. Asterisks denote statistical significance from a two-sided permutation test ($p < 0.01$, Bonferroni corrected, 10,000 permutations).

frontal cortices had lower SC-ISV than other frontal clusters, which could be due to unique cortical cytoarchitectonic divisions[41] that shape structural connections in those regions[11].

The ISV of MEG functional connectivity had similar spatial distributions in theta, alpha, and beta band oscillatory connectivity, in that frontal connections have higher ISV. Between the three lower frequency bands, the relative strength of the MEG-ISV in the visual, motor, and temporal cortices differed. The visual clusters had lower MEG-ISV than motor clusters in the alpha band. In the beta band, the motor clusters had low MEG-ISV, and temporal clusters had relatively higher MEG-ISV. Gamma-band MEG-ISV has a distinct spatial signature, possibly owing to its different connectivity patterns within and between large resting state networks[42]. By definition, regions with predominately strong and reliable connections would lead to smaller ISV than those with weak and variable connections. Hence, the frequency-dependent MEG-ISV patterns can also be related to different neural oscillators underpinning MEG functional connectivity at rest, which have been shown to reliably represent intrinsic fluctuations between spatially distant brain regions[43]. For example, alpha-band oscillation and oscillatory connectivity at rest predominately relate to synchronized neural activity in the occipital and temporal cortices[44]. In contrast, beta-band oscillation is the dominant feature of the sensorimotor system and is prominent in interhemispheric connections of motor cortices[45].

By using 7 T quantitative R1 imaging and multiple white matter compartment models, we reported the relationship between ISVs of brain connectivity and tissue microstructure. Quantitative R1 maps provide a good proxy of cortical myelin content[23]. Consistent with previous research, we observed high R1 values in the somatosensory, motor, auditory and visual cortices and low R1 values in the association cortices including frontal, parietal and temporal areas[24].

Both Cohort 1 and Cam-CAN data showed that SC-ISV negatively correlates with the R1 map. For MEG, the ISV in the beta band showed a similar negative correlation. This leads to two potential interpretations of the robust R1-ISV associations. (1) Both ISV and R1 maps follow the principal gradient of cortical hierarchy, and it is an emergent property of large-scale topography of the human brain. (2) Intracortical myelination does directly impact the variability of connectivity, and hence the spatial heterogeneity of ISV accompanies the regional difference of intracortical myelination, which is sensitive to in vivo MR contrasts such as R1 or T1w/T2w ratio.

Although these two propositions are not mutually exclusive, there is evidence supporting a direct impact of intracortical myelination on connectivity variability. Myeloarchitectonic-defined cortical regions have distinct functional, neurobiological and neurochemical properties[46]. Lightly myelinated areas are responsible for higher cognitive functions and become myelinated later in life[47]. These brain regions have lower neuronal density, large dendritic arborization, more spine density and more synapses thus more complex intracortical circuitry[12,48]. On the other hand, heavily myelinated cortical regions are thinner with a larger number of smaller cells and simpler dendritic trees[12]. Myelin-related factors reduce synaptic plasticity by inhibiting neurite growth such as new axonal growth and synapse formation[20]. Furthermore, lightly myelinated frontal and parietal regions require higher aerobic glycolysis than heavily myelinated areas[12]. The combination of low myelin content and high aerobic glycolysis may be a characteristic neurobiological feature of the association cortex, enabling adaptable and plastic neural circuitry[12], which in turn leads to high ISV in structural and functional connectivity.

The current study harnessed the sensitivity to the intra-axonal diffusion signal provided by the high b-values in the DWI acquisition. Both tissue-compartment models (NODDI and CHARMED) used here were validated by postmortem[49] or bio-mimetic phantom imaging[50]. They provide inferences to tissue microstructure unspecific to the DTI model, such as local fibre architecture, axonal morphology, and white matter myelin content. The PCA of DWI-based microstructural metrics yielded two biologically interpretable components explaining >90% of the variance, confirming recent results using a similar approach[26]. The alpha-to-gamma band MEG-ISV of a cortical region relates to the hindrance (i.e., the first principal component of tissue microstructure metrics) of the white-matter pathways originating from that region. Therefore, the ISV of functional connectivity not only depends on cortical myelin content but is also associated with white-matter tissue microstructure. Interestingly, the PC1's correlations with alpha/beta- and gamma-band MEG-ISV were in opposite directions. Alpha/beta- and gamma-band connectivity have selective sensitivity to short- and long-range regional coupling[51], whereas the balance between local and distant structural connections also varies between regions[52,53]. Further work is needed to examine the role of fibre length in frequency-specific MEG-ISV.

Three issues require further consideration. First, the choice and resolution of the brain atlas are important in estimating whole-brain connectivity and its variability[54]. We addressed this challenge by adapting all analyses based on the HCP-MMP atlas[13]. The original HCP-MMP atlas has a considerable size difference between ROIs. To minimise its impact on the calculation of the ISV, we upsampled the HCP-MMP atlas for the calculation of SC-ISV to have regions with similar sizes. For the calculation of the MEG-ISV, due to the limited spatial resolution and the rank of MEG data, we used a downsampled HCP-MMP atlas optimized for the MEG source-level signal-to-noise ratio[17]. Our approach enables cross-modal analyses on the same HCP-MMP cluster level. Future work could examine the ISV of connectivity using other brain atlases with various levels of granularity.

Second, imaging data acquired from the same participant would vary across multiple sessions. Such intra-subject, or inter-session, variability can be due to participant movement or equipment noise. For MEG-ISV, we employed two-stage artifact correction: (1) at the sensor level, epochs with artifacts were discarded; and (2) at the source level, head motion trace recorded from three gradiometers was regressed out from source reconstructed signals[55]. The pre-processed MEG data were further corrected for intra-subject variability in the calculation of MEG-ISV. For SC-ISV, the head motion was corrected during pre-processing, but intra-subject variability was not completely removed due to the limited availability of multi-session DWI data. Nevertheless, most of our analyses are between imaging modalities or between independent cohorts (e.g., between Cohort 1 and Cam-CAN). Therefore, results in the current study cannot be attributed solely to intra-subject variability or other measurement noise.

Third, an important next step is to link the cohort-based variability of brain connectivity to that of demographical and neurological or neuropsychiatric variables. The current study only included healthy participants with a narrow age range. Even in a homogeneous group, we observed substantial ISV changes across the brain in both functional and structural connectivity. Hence, our results are not confounded by age. A future direction would be to quantify how the spatial distribution of the connectivity ISV varies with age or during development, both of which have been shown to influence brain connectome[33,56]. The full Cam-CAN dataset provides an ideal opportunity for such analyses, as it contains rich imaging data across a large age span[19].

In healthy individuals, the ISV of structural and MEG functional connectivity exhibited spatial heterogeneity across the

cortex. The spatial patterns of ISVs were preserved across modalities in selective cortical clusters, and they relate to cortical myelin content as well as white-matter tissue microstructure. Our results define connectivity variability as an important cohort-level measure with strong neurobiological relationships. These findings further highlighted the characteristic features of individual differences in large-scale networks in the human brain.

## Methods

### Participants

*Study-specific participants.* 86 healthy participants were recruited from a local participant panel consisting of undergraduate and postgraduate students. Cohort 1 included 51 participants (36 females, age range 18-28 years, mean age 21.20 ± 2.74 SD years). All participants in Cohort 1 underwent a 3 T MRI session, and 28 participants in Cohort 1 completed two further MEG sessions. Cohort 2 included 35 participants (29 females, age range 18-35 years; mean age 21.22 ± 3.5 years), and all completed a 7 T MRI session. There was no significant difference in age (t(84) = −0.05, p = 0.96) or gender ($\chi^2$ = 1.69, p = 0.19) between the two cohorts. No participant reported a history of neurological or psychiatric illness. The study was approved by the Cardiff University School of Psychology Ethics Committee. All participants gave written informed consent. The use of two independent groups supports that our results on the relationship between the variability of brain connectivity and grey-matter microstructure are generalisable, rather than specific characteristics of a single group.

*Open-access dataset.* We used imaging data from the Cambridge Centre for Ageing and Neuroscience (Cam-CAN, https://www.cam-can.org). Cam-CAN is a large-scale population-derived cohort[19,57]. We chose all 50 participants between 20-30 years old from the Cam-CAN repository that have DWI data available (26 females, mean age: 25.78 ± 2.66 years). The inclusion of the Cam-CAN dataset enabled us to assess the replicability of our results from a different cohort, MR system and imaging sequence.

### MRI data acquisition for Cohort 1

Whole-brain, multi-shell, diffusion-weighted images (DWI) were acquired from all participants in Cohort 1, using a Siemens 3 T Connectom MRI scanner with a gradient of 300 mT/m (Siemens Medical Systems). The superior gradient performance of the Connectome scanner compared to conventional MR systems enables DWI acquisition at high diffusion weightings.

The spin-echo echo-planar imaging (EPI) pulse sequence used a high angular resolution DWI protocol (echo time 59 ms, repetition time 3000 ms, voxel size 2 × 2 × 2 mm). Diffusion sensitizing gradients were applied in 20 isotropic directions at b-values of 200 and 500 s/mm², in 30 isotropic directions at a b-value of 1200 s/mm² and in 61 isotropic directions at b-values of 2400, 4000, 6000 s/mm². Thirteen volumes with no diffusion weighting (b = 0 s/mm²) interleaved across the sequence were also acquired. To correct for susceptibility-induced distortions, three images at b = 0 s/mm² and 30 diffusion directions at b = 1200 s/mm² were acquired with the opposite phase encoding direction. Participants also underwent high-resolution T1-weighted magnetization prepared rapid gradient echo scanning (MPRAGE: echo time 3.06 ms; repetition time 2250 ms, flip angle 9°, the field of view=256 × 256 mm, voxel size 1 × 1 × 1 mm).

### MEG data acquisition for Cohort 1

Whole-head MEG recordings were acquired in a magnetically shielded chamber, using a 275-channel CTF radial gradiometer system (CTF Systems, Canada) at a sampling rate of 1200 Hz. One sensor was turned off during recording due to excessive noise. Additional 29 reference channels were recorded for noise cancellations and the primary sensors were analyzed as synthetic third-order gradiometers. Continuous horizontal and vertical bipolar electro oculograms (EOG) were recorded to monitor blinks and eye movements. Participants were seated comfortably in the MEG chair and their head was supported with a chin rest to minimise head movement. For MEG/MRI co-registration, the head shape with the position of coils was digitized using a Polhemus FASTRAK (Colchester, Vermont). Participants were instructed to rest with their eyes open and fixate on a red dot with a grey background, presented through a back projector. Each recording session lasted approximately 8 minutes. 28 participants in Cohort 1 underwent two resting-state MEG sessions on different days (1 to 8 days between two sessions). For all MEG analyses, we therefore used data from those 28 participants. The MEG dataset was used in previous studies for different analyses[17,58,59].

### Cortical segmentation and reconstruction

Freesurfer (version 5.3.0, http://surfer.nmr.mgh.harvard.edu) was used to process T1-weighted MPRAGE images, including motion correction, intensity normalization, skull-stripping, white-matter segmentation, tessellation, smoothing, inflating and spherical registration[60]. After pre-processing, the surface of the grey matter/white matter boundary was generated, together with inner skull, scalp and pial images. Conformed and intensity normalized T1-weighted image was registered to the mean non-diffusion image (b = 0 s/mm²) by using a boundary-based rigid body registration with six degrees

of freedom. For each participant, the forward and inverse transformation matrices between the native DWI space and T1 space were used for subsequent co-registration and tractography analyses.

### Cortical parcellation

We parcellated the cortex into regions of interest (ROIs) with different spatial resolutions based on the Human Connectome Project Multi-Modal Parcellation (HCP-MMP) atlas[13]. The original HCP-MMP atlas includes 358 ROIs (excluding hippocampal parcellations), and the surface size varied substantially between those ROIs (from 122 mm² to 3198 mm²). The large size difference between ROIs may affect subsequent inter-subject variability analyses. To reduce this potential confounding effect, for analyses of MRI and DWI data, we upsampled the original HCP-MMP atlas with more, smaller ROIs, by using the mris_divide_parcellation function in Freesurfer. Each ROI of the original HCP-MMP atlas was subdivided perpendicular to the long axis of the ROI, with new subdivided ROIs to have a reduced variability of surface size. This process was conducted for all ROIs of the original HCP-MMP atlas, resulting in 664 ROIs. We refer to this high-resolution atlas as the upsampled HCP-MMP atlas.

For analyses of MEG data, we used a downsampled version of the HCP-MMP atlas for MEG source reconstruction. The downsampled HCP-MMP atlas contains 230 ROIs and is optimized to match the spatial resolution and the rank of MEG data[17].

In both upsampled and downsampled HCP-MMP atlas, the categorization of 22 cortical clusters in the original HCP-MMP atlas was maintained (Supplementary Fig. 2). Therefore, our modal-dependent parcellation procedure allows us to conduct analyses between imaging modalities at the cluster level.

### DWI data pre-processing

DWI data were converted from DICOM to NIfTI format using dcm2nii. For each participant, the images were skull-stripped using FSL (version 6.0.1, https://fsl.fmrib.ox.ac.uk) and denoised using the MP-PCA noise estimation function in MRTrix[61] (version 3, https://www.mrtrix.org). Following drift correction[62], images were corrected for susceptibility-induced distortions, eddy currents and head motion using FSL. After correcting for gradient nonlinearity and Gibbs ringing artefacts[63], the mean non-diffusion image was obtained by average across all images with zero b values. Fibre Orientation Distribution Functions (fODFs) were derived from multi-shell multi-tissue Constrained Spherical Deconvolution[64]. The fODFs were then normalized with the mtnormalise tool from MRTrix to enable multisubject comparison.

### Tractography

A probabilistic tractography algorithm based on the second-order integration over fODFs was used[65] with the anatomically-constrained tractography (ACT) framework in MRTrix[66] (fibre orientation distribution amplitude threshold 0.1, step size 1 mm, 4 samples per step, maximum curvature per step 45°, the cut-off value for terminating tracks 0.06, minimum track length 5 mm, the maximum tract length 300 mm, maximum number of streamlines 10 million). The whole-brain tractography procedure used the grey matter/white matter boundary obtained from Freesurfer as the seed mask. Segmented tissue maps were used to constrain tractography.

ACT discarded streamlines that did not reach the target mask and were not anatomically plausible[66]. To further reduce outlier streamlines, we applied a streamline trimming procedure based on geometric clustering[67]. For each pair of ROIs, we sampled all streamlines connecting the two ROIs along their length. Streamlines were then grouped into clusters according to their Euclidean distance from each other. New clusters were formed if a streamline was more distant than a predetermined threshold that was set as 20 mm. After clustering, clusters with less than 3 streamlines were considered outliers and were discarded (Fig. 1a and Supplementary Fig. 6). This procedure removed 2.58% ± 3.91% of outlier streamlines across all participants.

### DWI-based structural connectome

For each participant, we generated a structural connectivity matrix based on the upsampled HCP-MMP atlas. We counted the number of streamlines connecting each pair of ROIs. This step resulted in an ROI-by-ROI matrix of streamline counts. The matrix was then thresholded to have a minimum of 50 streamlines. ROIs with streamlines less than this threshold were considered unconnected. The final structural connectivity matrix was obtained by normalizing each row of the streamline count matrix. Hence, values in the structural connectivity matrix represent the connection probability from one ROI to the rest of the ROIs.

### White-matter microstructural measures

We fitted three microstructural models to the DWI data: the conventional DTI model, the neurite orientation dispersion and density imaging (NODDI) model[25], and the composite hindered and restricted model of diffusion (CHARMED)[68] (Fig. 1a).

From the pre-processed DWI data, we calculated fractional anisotropy (FA), mean diffusivity (MD), radial diffusivity (RD) and angular diffusivity (AD) by fitting the b = 0, 500 and 1200 s/mm² shell data to the DTI model. This step used the MRTrix functions dwi2tensor and tensor2metric. We used data from lower b values since the DTI model is based on hindered diffusion in the extra-axonal space which is more sensitive to lower b values[69].

The DWI data from all available b values were fitted to the NODDI model using the NODDI MATLAB Toolbox v1.0.1 to calculate the volume fraction of the intracellular compartment (ICVF) and orientation dispersion index (ODI). The NODDI model includes three compartments: intracellular space, extracellular space, and CSF[25]. In this way, restricted diffusion perpendicular to neurites and unhindered diffusion along them is explicitly modelled. ICVF quantifies the volume of the compartment that contains the axons and dendrites, whereas ODI represents angular variation in neurite orientation.

We calculated restricted diffusion signal fraction maps (FR) by fitting the CHARMED model[68] to the whole DWI data, which characterizes white matter in restricted and hindered compartments. CHARMED is more sensitive to local fibre orientation than the standard DTI model, thus giving a better estimate of restricted diffusion of intra-axonal water molecules.

**Connectome microstructure**. We calculated 7 voxel-wise microstructural measures including four from the DTI model (FA, MD, RD, and AD), two from the NODDI model (IVCF and ODI) and one from the CHARMED model (FR). All microstructural measures were sampled along the streamlines by using tcksample function from MRTrix[61]. We took the median value of measures along and across streamlines to characterize microstructural properties of region-to-region connections. In this way, we obtained a $664 \times 664 \times 7$ matrix for each participant, representing microstructural measures of structural connectivity. We converted microstructural measures to z-score per subject to avoid scale differences between measures and participants.

Since DWI-based microstructural measures contain mutually complementary information[26,70], we reduced the dimensionality of the microstructural measures. This was achieved by applying PCA to the seven microstructural measures across participants and structural connections. Similar to previous results[26], we focused on the first two principal components, which in total explained >90% of the variance in the data (Supplementary Fig. 5).

**Cam-CAN DWI data and analyses**. All Cam-CAN participants included in the current study had DWI data acquired from a Siemens 3 T MAGNETOM scanner. A spin-echo EPI sequence was used (echo time 104 ms, repetition time 9100 ms, voxel size $2 \times 2 \times 2$ mm). Diffusion-sensitizing gradients were applied in 30 isotropic directions at b-values of 1000 and 2000 s/mm². The participants also underwent a T1-weighted scan (MPRAGE: echo time 2.99 ms; repetition time 2250 ms, flip angle 9°, the field of view = $256 \times 240$ mm, voxel size $1 \times 1 \times 1$ mm).

We used the Cam-CAN dataset to assess whether the pattern of inter-subject variability of structure connectivity is replicable. Hence, the analysis steps of the Cam-CAN DWI data were as close as possible to that of the Cohort 1 data. Preprocessing included denoise, correction for Gibbs ringing artefacts, eddy currents and head motion. Susceptibility correction was not performed because there was no DWI data with the opposite phase encoding direction. We used the same analysis pipeline described above on the T1 structural image for cortical segmentation, reconstruction and parcellation. Pre-processed DWI and T1-weighted data entered the same pipeline for whole-brain tractography and calculating structural connectivity. Because of the relatively low b-values in the Cam-CAN DWI sequence, fitting microstructural models such as CHARMED is sub-optimal. Hence, we did not measure microstructural metrics in the Cam-CAN data.

**MEG pre-processing**. MEG data is pre-processed following an analysis pipeline[17] (Supplementary Fig. 1). Continuous raw MEG data was imported to Fieldtrip[71], downsampled to 256 Hz, bandpass filtered at 1-100 Hz (4th order two-pass Butterworth filter). Data was subsequently notch-filtered at 50 and 100 Hz to remove line noise. Visual and cardiac artifacts were removed using ICA decomposition. Identification of visual artifacts was aided by simultaneous EOG recordings. Between two and six components were removed for each subject. After segmenting time courses into 2-second epochs, an automated artifact detection was applied using the Fieldtrip functions ft_artifact_clip, ft_artifact_jump, and ft_artifact_z-score. Trials with artifacts were rejected from further analysis.

**MEG source reconstruction**. The inner skull, scalp, pial, and grey matter/white matter boundary surfaces generated from Freesurfer and the MRI scan were imported into the Brainstorm software[72]. An automated procedure was used to align these data to the MNI coordinate system. The midpoint between the pial surface and grey matter/white matter boundary was extracted and downsampled to approximately 10,000 homogeneously spaced vertices to generate a cortical surface of dipole locations using the iso2mesh software implemented in Brainstorm. The inner skull surface was similarly downsampled to 500 vertices. These surfaces were then exported to Matlab, where the scalp surface was used to align the structural data with the MEG digitizers. The aligned MEG gradiometers, inner skull surface, and cortical surface were then used to construct a realistic, subject-specific, single-shell forward model. Dipole orientations were fixed normal to the cortical surface.

Exact low-resolution electromagnetic tomography (eLORETA) was used to reconstruct source activity[73]. eLORETA is a linear, regularized, weighted minimum-norm inverse solution with exact, zero error localization[73], which has previously been shown to perform well with MEG resting-state data[17] and is suited to study of whole brain synchronization[74].

Head position was estimated from the circumcentre of three head localization coils in each trial. Head movement trajectories that contain transformations and rotations were z-transformed and regressed out from source level MEG time series[55,75].

The cortical surface was aligned to the MEG-optimized, downsampled version[17] of the HCP-MMP atlas (115 cortical ROIs per hemisphere) in Freesurfer. The time series of each ROI was calculated as the time course of the first principal component of all voxels within the ROI.

**MEG-based functional connectome**. Functional networks were constructed using amplitude envelope correlation (AEC) within four frequency bands (theta 4-8 Hz, alpha 8-13 Hz, beta 13-30 Hz, and gamma 30-100 Hz). For a given frequency band, data were bandpass filtered. We performed leakage correction using multivariate orthogonalization[76], then computed the amplitude envelope (which was low-pass filtered at 1 Hz and downsampled to 0.5 Hz) and calculated the correlation between pairs of ROIs to construct functional networks. This metric was chosen to quantify amplitude relationships due to its high reliability[76].

MEG functional connectivity matrices were thresholded based on a graph metric to balance between integrated and segregated networks[77]. We compared different connection density values with respect to cost-efficiency, which is defined as the difference between global efficiency and topological cost expressed as the density of the network. We calculated the density at 0.25 based on this metric for MEG connectivity matrices at all frequencies and thresholded MEG connectivity matrices to retain the strongest 75% connections.

**7 T MRI data acquisition for Cohort 2**. Whole-brain, high-resolution and high-field structural imaging were acquired from all participants in Cohort 2 on a Siemens 7 T Magnetom MRI scanner (Siemens Medical Systems, Germany) using a 32-channel head coil (Nova Medical, USA). The MP2RAGE sequence was used[78], which included two MPRAGE acquisitions with different flip angles and inversion times (echo time 2.68 ms, repetition time 6000 ms, first inversion time 800 ms, second inversion time 2700 ms, first flip angle 7°, second flip angle 5°, voxel size $0.65 \times 0.65 \times 0.65$ mm³). To correct for the RF transmit field $B_1^+$, whole-brain $B_1^+$ images were acquired using the saturation-prepared with 2 rapid gradient echoes (SA2RAGE) sequence[79] (echo time 1.16 ms, repetition time 2400 ms, first inversion time 540 ms, second inversion time 1800 ms, first flip angle 4°, second flip angle 11°, voxel size $3.25 \times 3.25 \times 3$ mm³).

**7 T MRI data preprocessing and quantitative R1 mapping for Cohort 2**. The MP2RAGE sequence generated two images at the first (INV1) and second (INV2) inversion times. For each participant, after an online linear interpolation procedure on INV1 and INV2 maps[78], we obtained a quantitative T1 map that was corrected from proton density contrast, T2* contrast and RF receive field $B_1^+$. By combining normalized INV1 and INV2 images, we also obtained a T1-weighted image from the same sequence.

The $B_1^+$ map from the SA2RAGE was registered to the individual participant's INV2 map using the linear registration function FLIRT in FSL[80]. The co-registered $B_1^+$ map was then used to correct residual $B_1$ inhomogeneities in the T1-weighted and T1 maps. Because the INV2 image has a better contrast between brain tissues and the skull, we used the brain extraction function BET in FSL to obtain a skull-stripped brain mark from the INV2 image. Next, the Java Image Science Toolkit[81] and the CBS High-Res Brain Processing tools of the MIPAV platform[82] took the T1-weighted image and the brain mask as inputs and generated probabilistic maps of the intracranial dura and arteries. These probabilistic maps were thresholded and used as a mask to remove the dura and arteries in the T1 and T1-weighted images. Finally, for each participant, we calculated a quantitative R1 map (i.e., the longitudinal relaxation rate, R1 = 1/T1) from the bias-corrected, brain-extracted T1 image, which is sensitive to cortical myelin content[24]. Two participants had distorted R1 images and were removed from subsequent analyses. A threshold of maximum voxel intensity was applied to the R1 map to discard voxels with large artifacts. This step resulted in the loss of R1 estimates in the lateral temporal cluster, because those voxels are commonly affected by the amplified background noise in the MP2RAGE images[83] that can result in poor skull stripping and brain segmentation.

**Cortical reconstruction and parcellation of R1 maps for Cohort 2**. We projected T1w images on the cortical surface by using recon-all from Freesurfer in two steps by skipping the skull stripping with hires option to keep the original high resolution of T1w images. We registered the upsampled HCP-MMP atlas onto the native T1w space for each participant. Because R1 values vary from the white matter to the pial surface[23], the submillimetre resolution of our data allows us to obtain R1 maps from the middle depth of the cortex using mri_vol2surf (i.e., at the 50% depth between white matter and pial surface). The R1 value of each ROI was then calculated by averaging the R1 values across all vertices in the ROI.

**Inter-subject variability of brain connectivity**. The inter-subject variability (ISV) is calculated for the connectome of each imaging modality (DWI and MEG, Fig. 1b). Consider a connectivity matrix $S_{(R \times R)}^i$ of subject $i$, where $R$ denotes the

total number of ROIs. We first calculated a similarity metric $Dist_{subj}(p, r)$ for each ROI $r$ between any pair of subjects $p$ as follows:

$$Dist_{subj}(p, r) = CD[S^m(r, :), S^n(r, :)],$$
$$\forall p = (m, n),\ m, n = 1, 2, \ldots, N; m \neq n; r = 1, 2, \ldots, R \quad (1)$$

where $S^m(r, :)$ denotes the $r$th row of $m$th subject's connectivity matrix and $N$ denotes the total number of subjects. We used the cosine distance (CD) as the similarity metric. The CD between two vectors $x, y \in \mathbb{R}^{1 \times R}$ is defined as:

$$CD(x, y) = \frac{\sum_i x(r)y(r)}{\sqrt{\sum_r x(r)^2}\sqrt{\sum_r y(r)^2}},\ r = 1, 2, \ldots, R. \quad (2)$$

The ISV of each ROI $r$ is then calculated by averaging the similarity measure across all subject pairs:

$$ISV(r) = \frac{1}{\binom{N}{2}}\sum_p Dist_{subj}(p, r). \quad (3)$$

To eliminate the residual effect of ROI size on ISV, we regressed out the ROI size (averaged across subjects) from the ISV vector. For each cortical cluster defined in the HCP-MMP atlas[13], we then calculated the cluster-level ISV by averaging across the ROI-level ISV measures within each cluster. The variance of ISVs was calculated from 5000 bootstrap samples on $Dist_{subj}$ matrices.

**Inter-subject variability corrected for intersession variability.** Two sessions of MEG resting-state data were acquired from each participant. For the ISV of MEG functional connectivity, we further corrected for within-subject, inter-session variability (Supplementary Fig. 7). This step removed confounding effects of measurement noise.

For each subject $m$, consider the MEG connectivity matrices are $S^m_{1,(R \times R)}$ and $S^m_{2,(R \times R)}$ from two sessions. We first calculated the the similarity metric $Dist_{ses}(m, r)$ between the two sessions for each ROI $r$, serving as a measure of intersession variability for the subject $m$.

$$Dist_{ses}(m, r) = CD[S^m_1(r, :), S^m_2(r, :)],\ m = 1, \ldots, N; r = 1, \ldots, R. \quad (4)$$

For each pair $p$ of subjects $m$ and $n$, the intersession variability of the two subjects was then regressed out from the inter-subject similarity metric of the subject pair $p$:

$$Dist_{subj-corrected}(p, :) = Dist_{subj}(p, :) - \beta_1 \cdot \frac{1}{2}\left(Dist_{ses}(m, :) + Dist_{ses}(n, :)\right) - \beta_0$$
$$\forall p = (m, n), m, n = 1, 2, \ldots, N; m \neq n, \quad (5)$$

where $\beta_1$ and $\beta_0$ are the slope and intercept from linear regression, respectively. The ISV of each ROI $r$ for MEG functional connectivity is then calculated by averaging the corrected similarity measures across all subject pairs.

$$ISV_{corrected}(r) = \frac{1}{\binom{N}{2}}\sum_{\substack{p=(m,n) \\ m \neq n}} Dist_{subj-corrected}(p, r). \quad (6)$$

The ISV corrected for intersession variability was first calculated separately for two sessions. We then averaged the ISV measures from two MEG sessions to obtain the final values.

**Cross-modal analyses of inter-subject variability.** We conducted two analyses on the inter-subject variability of connectivity between imaging modalities. First, previous studies showed no strong relationship between the ISVs of structural and functional connectivity[18]. The current study examined a different question: for a given cortical cluster, do a pair of participants with a higher structural connectivity difference also exhibit a higher MEG functional connectivity difference? This analysis allows us to examine the alignment of inter-subject differences in brain connectivity at the cortical cluster level. In this analysis, we regressed out the ROI size from each row of the $Dist_{subj}$ matrix in Eq. 1 to minimize the impact of ROI size. For each macroscopic cortical cluster defined in the HCP-MMP atlas, we then averaged the similarity metrics across all ROIs within the cluster for each pair of subjects. For each cortical cluster and each imaging modality, this calculation yielded a vector of similarity metrics from all subject pairs. We calculated the Spearman rank correlation coefficients between similarity vectors from different imaging modalities, testing the hypothesis that participants with more distinct connectivity patterns in one imaging modality also differ more in their connectivity patterns in another modality. To correct for multiple comparisons across the 22 cortical clusters, we used permutation-based maximum statistic[84] with 10,000 permutations to control for the family-wise error (FWE) rate.

Second, we examined to what extent the ISV of brain connectivity is associated with the R1 value (from 7 T MP2RAGE data in cohort 2) and white-matter microstructure (from DTI, NODDI and CHARMED models fitted to the DWI data in cohort 1). For this analysis, Spearman rank correlations were calculated across all ROIs between the ISV of connectivity and tissue microstructure measures. For calculations involving MEG-ISV, the R1 map and the white-matter microstructure maps were downsampled to the MEG-optimized atlas to match the spatial resolution of MEG-ISV.

**Correction for spatial autocorrelation.** Most brain maps contain inherent spatial autocorrelation: spatially closer regions tend to have more similar values. When comparing brain maps, their spatial autocorrelation can inflate statistical results because values across ROIs are not independent, leading to increased type I error[85]. In the current study, for each correlational analysis of brain maps, we used Brainsmash[86] to generate 5000 surrogate brain maps with the same spatial auto-correlation as in the observed data. We used geodesic distance (i.e., the distance along the cortical surface) between ROIs to quantify spatial autocorrelation in observed and surrogate brain maps. The surrogate maps were then served as a null distribution to calculate corrected two-sided permutation p-values for the statistical tests of spatial correspondence between brain maps. The corrected p-values reflect a test against a stringent null hypothesis that accounts for the ubiquitous spatial autocorrelation of brain maps. We used the notation SA-corrected when reporting corrected p-values.

**The multiple demand network.** The multiple demand network (MD) includes brain regions with integrative properties in response to cognitively demanding tasks[27]. We used the most recent definition of an extended MD network based on common BOLD functional MRI responses in three cognitive domains (working memory, math/language and reasoning)[28]. In the HCP-MMP atlas, the extended MD network was defined by a core of 10 ROIs and a penumbra of 17 ROIs per hemisphere, primarily in the frontoparietal cortex. To facilitate comparisons, we referred to non-MD regions as cortical ROIs that are not core MD or penumbra regions.

**Statistics and reproducibility.** ISV calculations, cross-modal alignments and permutation tests were conducted using Matlab (version R2018b). Corrections for spatial autocorrelation were conducted using the Brainsmash package[86] in Python (version 0.11.0, Python version 3.8.8). Brain maps were rendered using the Matlab GIfTI library (version 2.0).

**Reporting Summary.** Further information on research design is available in the Nature Research Reporting Summary linked to this article.

## Data availability

DWI, MEG and 7 T MRI data that support the findings of this study and data from all analyses are available in OSF with the unique identifier [https://doi.org/10.17605/osf.io/rqj8a]. A part of data used in the preparation of this work were obtained from the Cam-CAN repository (https://www.cam-can.org).

## Code availability

The scripts that support the calculation of the ISV and statistical analyses used in this study are available in OSF with the unique identifier [https://doi.org/10.17605/osf.io/rqj8a].

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

## Acknowledgements

This study was supported by the European Research Council (716321), the UK Medical Research Council (MR/N01233X/1) and a Wellcome Trust Institutional Strategic Support Fund (ISSF). This research was funded in part by the Wellcome Trust (104943/Z/14/Z). R.S. was supported by a PhD studentship from the China Scholarship Council. A.Ö. was supported by a PhD studentship from the Turkish Ministry of National Education. M.J.S. was supported by a PhD studentship from Cardiff University School of Psychology. We thank Slawomir Kusmia for MRI support, Greg Parker for DWI analysis support, and Krish Singh for helpful comments. This study used data provided by the Cambridge Centre for Ageing and Neuroscience (Cam-CAN). Cam-CAN funding was provided by the UK Biotechnology and Biological Sciences Research Council (BB/H008217/1), together with support from the UK Medical Research Council and the University of Cambridge, UK. For the purpose of Open Access, the authors have applied a CC BY public copyright licence to any Author Accepted Manuscript version arising from this submission.

## Author contributions

E.K. and J.Z. conceived the project. R.S., A.Ö., M.J.S., and J.Z. collected imaging data. E.K., L.T., R.S., and J.Z. analysed the data. E.K. and J.Z. wrote the initial draft of the manuscript. A.D.L. and K.S.G. contributed to the editing and review of the manuscript.

## Competing interests

The authors declare no competing interests.
