## [Peer Review File · Communications Biology]

Reviewers' comments:

Reviewer #1 (Remarks to the Author):

In this study, the authors use a multimodal imaging approach to map interindividual variability using diffusion-weighted imaging (DWI) with probabilistic tractography, BOLD-fMRI, and MEG.

The participants need to be better described. 2.1 details two cohorts but no mention of the rationale for why there are two cohorts or why one group was scanned on a 3T and the other on a 7T scanner. In addition, the used multimodal imagings which in group sizes of 29 and 35 warrants the question about power and effect sizes. It is not specified where the participants were recruited from but given the average age of 20 years it seems it might be a classical student sample, which opens up queries about the generalisability of these findings to other ages.

For cohort 1, the authors opted for probabilistic tractography with a clustering algorithm and additionally fitted 3 microstructural models to the DWI data (DTI/ CHARMED/NODDI). What was the rationale to use all three of them in the first place and which software was used to fit the DTI? The tractography results are based on a methodological approach - rather than anatomical - to structural connectivity. How can the authors be sure to have included genuine connections rather than artefacts?

How does the interindividual variability compare to intra-individual variability? Is it possible that the variability between people is comparable to the variability of the same brain scanned at different time points?

How do the results of structural and functional connectomes showing lower variability in sensory and visual cortices sit in comparison to the previous literature that showed high variability in these regions? Some references that come to mind:

Yousry TA, Schmid UD, Alkadhi H, et al. Localization of the motor hand area to a knob on the precentral gyrus. A new landmark. *Brain*. 1997;120 (Pt 1):141-157. doi:10.1093/brain/120.1.141

Uylings HBM, Rajkowska G, Sanz-Arigita E, Amunts K, Zilles K. Consequences of large interindividual variability for human brain atlases: converging macroscopical imaging and microscopical neuroanatomy. *Anat Embryol* . 2005;210(5-6):423-431. doi:10.1007/s00429-005-0042-4

Geyer S, Schleicher A, Zilles K. Areas 3a, 3b, and 1 of Human Primary Somatosensory Cortex: 1. Microstructural Organization and Interindividual Variability. *Neuroimage*. 1999;10(1):63-83. doi:10.1006/nimg.1999.0440

Croxson PL, Forkel SJ, Cerliani L, Thiebaut de Schotten M. Structural Variability Across the Primate Brain: A Cross-Species Comparison. *Cereb Cortex*. 2018;28(11):3829-3841. doi:10.1093/cercor/bhx244

Finally, in the absence of any cognitive and clinical data, how can the authors be sure these results are meaningful beyond the immediate study?

Reviewer #2 (Remarks to the Author):

Thank you for the opportunity to review "The inter-individual variability of connectome maintains regional-selective consistency between modalities and covaries with 3 tissue microstructure". The authors compute inter-subject variability for three imaging modalities (DWI, fMRI and MEG). They find that the patterns are heterogeneous and that they are often anticorrelated with local R1 measurements.

The study certainly addresses an interesting question about cross-modal correspondence depends on inter-individual variability. The data processing and most of the analyses are carried out at a high technical standard. At the same time, there a number of conceptual and methodological points that should be addressed.

(1) It is not clear what the overall conclusion is. What do the authors make of the differing patterns of ISVs? Why should areas with greater R1 have lower ISVs? There is a notable lack of conceptual coherence.

(2) Can you just merge Figures 2-4? Also label colormaps for panels a, and provide more info for panels b.

(3) Throughout the manuscript, the parcellation does not look like MMP – it looks like it has substantially fewer regions. Can the authors confirm?

(4) I don't think that it is sufficient to just show alpha and beta based on one prior paper. Please show results for all the canonical frequency bands, from delta to gamma.

(5) The labels in the bars have acronyms that are a bit mysterious. For examples, what is FRN?

(6) When comparing ISVs, why not correlate the ISV maps instead?

(7) Combine Figs 5 and 6. Also missing caption for Figure 6.

(8) Figure 7: please use spatial autocorrelation -preserving null models to assess the significance of the correlation:

Alexander-Bloch, A. F., Shou, H., Liu, S., Satterthwaite, T. D., Glahn, D. C., Shinohara, R. T., ... & Raznahan, A. (2018). On testing for spatial correspondence between maps of human brain structure and function. *Neuroimage*, 178, 540-551.

Burt, J. B., Helmer, M., Shinn, M., Anticevic, A., & Murray, J. D. (2020). Generative modeling of brain maps with spatial autocorrelation. *NeuroImage*, 220, 117038.

Markello, R. D., & Misic, B. (2021). Comparing spatial null models for brain maps. *NeuroImage*, 236, 118052.

Dear Reviewers,

Thank you for your comments on our manuscript [COMMSBIO-21-3067-T]. We are pleased that you have found our work of general interest and importance. The specific comments have been helpful and constructive in the substantial revision of the manuscript. We have addressed each of the issues raised. This Revision has included new analyses and new datasets.

We summarize below our response to your comments and the changes implemented in the revision. Together with this letter, we attached the edited version of the original manuscript with changes highlighted, as well as the revised manuscript with all changes applied. The line numbers indicated in the responses below refer to the revised manuscript with tracked changes highlighted.

The last author would also like to apologize for the delay in submitting the revision, which is due to his teaching commitments, staff changes in the research team, and the time needed for additional data collection and analyses.

Yours sincerely,

The corresponding author on behalf of the research team

Reviewer 1

1. The participants need to be better described. 2.1 details two cohorts but no mention of the rationale for why there are two cohorts or why one group was scanned on a 3T and the other on a 7T scanner. In addition, the used multimodal imagings which in group sizes of 29 and 35 warrants the question about power and effect sizes. It is not specified where the participants were recruited from but given the average age of 20 years it seems it might be a classical student sample, which opens up queries about the generalisability of these findings to other ages.

In the Revision, we have provided further description of the cohorts and justified our choices of scanners (lines 101, 105, 116 and 377).

Participants

Cohorts 1 and 2 were recruited from our School of Psychology participant panel. Most participants in the panel are undergraduate or postgraduate students. The use of two independent groups supports that our results on the relationship between the ISV of connectivity and R1 maps are generalisable, rather than specific characteristics of a single group (line 99).

The choice of scanners and sequences

We used two MR systems to maximize their unique advantages. First, Cohort 1 was scanned on a Siemens 3T Connectom MRI scanner. The Connectom scanner has superior gradient performance (300 mT/m) compared to conventional MR systems, enabling DWI acquisition at high diffusion weightings, e.g., b-values up to 6000 s/mm² in our sequence (Sotiropoulos et al., 2013; Jones et al., 2018; Fan et al., 2022). (line 116). The strong diffusion weighting has been shown to improve the reliability of tractography (Maffei et al., 2019) and the fit of microstructural models that were used to infer tissue microstructure in the current study (i.e., NODDI (Zhang et al., 2012) and CHARMED (Assaf and Basser, 2005)).

Second, from Cohort 2, we obtained quantitative R1 mapping using an MP2RAGE sequence on a Siemens 7T scanner. R1 measure is sensitive to the cortical myelin content, validated by post-mortem histological studies (Schmierer et al., 2004, 2008).

At 7T, MP2RAGE achieves whole-brain coverage with high spatial resolution (650 μm isotropic) under 10 minutes of acquisition time (Marques et al., 2010). This sequence minimizes the strong inhomogeneity of the B1 field at 7T and produces robust R1 maps with high reproducibility (O'Brien et al., 2014). We further corrected B1⁺ inhomogeneity using a SA2RAGE sequence (Eggenchwiler et al., 2012; Marques and Gruetter, 2013). Because R1 values vary from the white matter to the pial surface (Marques et al., 2017), the submillimetre resolution of our data allows us to obtain R1 maps from the middle layer of the cortex (line 377).

Generalisability

To further address the Reviewer's question on generalisability (see also Point 4 to Reviewer 1), we recruited further 22 participants from the same participant panel. These participants underwent a DWI session with the same MR sequence. In the Revision, we have reported whole brain tractography and the ISV of structural connectivity from a total of 51 participants (22 new participants and 29 in the original manuscript, see Figure panel A below). Our main conclusions remain unchanged.

To assess the reproducibility of our DWI results, we have additionally analysed the SC-ISV using a cross-sectional imaging dataset from the Cambridge Centre for Ageing and Neuroscience (Cam-CAN, Taylor et al., 2017). Here, we included the DWI results from all Cam-CAN participants between 20-30 years old (50 participants, 26 females, mean age: 25.78 ± 2.66 years), which have an age span similar to our Cohorts 1 and 2. It is worth noting that Cam-CAN data was acquired from a different scanner and a very different DWI sequence. Nevertheless, across the cortex, the SC-ISVs between our data and Cam-CAN were highly correlated ($R=0.76$, $p=0.002$ corrected for spatial autocorrelation, $p=9.67 \times 10^{-125}$ uncorrected, Spearman's correlation, see Figure panels B and C below). Furthermore, the SC-ISV from Cam-CAN negatively related to the R1 value ($R=-0.45$, $p=0.002$ corrected for spatial autocorrelation), replicating our results from Cohort 1. In the Revision, we have included the results from Cam-CAN dataset (lines 105 and 260 in Method, Figures 2 and 4).

Figure. SC-ISV related results. **A.** SC-ISV from Cohort 1 (51 participants). The brain maps (upper left) rendered the z-score of each ROI's ISV on the cortical surface. The barplot (right) showed the mean ISV of each of the 22 cortical clusters, which was estimated from all ROIs within each cluster. Error bars denote 95% confidence intervals. The matrix (lower left) denoted the group average of the structural connectome. **B.** SC-ISV from Cam-CAN young group (50 participants). **C.** Null distributions and observed Spearman correlation coefficients between Cohort 1 and Cam-CAN SC-ISV. SA-preserving null distributions (in red) were obtained from correlations with 5,000 surrogate maps with the same spatial autocorrelation as in the empirical brain maps using *brainsmash* (Burt et al., 2020). SA-independent null distributions (in blue) were obtained from correlations with 5,000 iterations of randomly shuffled values (** $p < 0.05$ SA-corrected). Dashed vertical line indicates the empirical Spearman's correlation coefficient (see also Point 9 to Reviewer 2 for details). **D.** Correlations between SC-ISV (left: Cohort 1; right: Cam-CAN) and R1 values across the cortex. The figure panels presented above are a part of the new Figures 2, 4 and Supplementary Figure S6 in the Revision.

2. For cohort 1, the authors opted for probabilistic tractography with a clustering algorithm and additionally fitted 3 microstructural models to the DWI data (DTI/CHARMED/NODDI). What was the rationale to use all three of them in the first place and which software was used to fit the DTI?

In the Revision, we have clarified that we used MRTrix3 to obtain the DTI metrics (line 230). We fitted a DTI model to pre-processed DWI data using the MRTrix function *dwi2tensor* and obtained DTI metrics using the function *tensor2metric*.

We have also highlighted the need to combine multiple microstructural models in the new Figure 4, which illustrates the interdependency between microstructural metrics. It is well acknowledged that the DTI model only provides limited insights to tissue microstructural properties, because it assumes a single tissue compartment and hence cannot distinguish between intracellular and extracellular space. On the other hand, metrics from more sophisticated microstructural models (e.g., NODDI and CHARMED) offer better biological validity, but they are not independent from DTI measures. Recent studies on children showed that the first two principal components from multiple microstructural metrics based on DTI, NODDI and CHARMED models explain >80% of the variance (Chamberland et al., 2019). In the current study, we have replicated this finding in an adult group. After PCA-based dimensionality reduction, we can relate two biologically relevant microstructural components to the ISV of structural and functional connectivity.

3. The tractography results are based on a methodological approach - rather than anatomical - to structural connectivity. How can the authors be sure to have included genuine connections rather than artefacts?

To address this issue, we have used an updated and more efficient tractography procedure in the Revision (line 191). More specifically, we used the iFOD2 algorithm (second-order integration over fiber orientation distributions) with anatomically-constrained tractography (ACT) framework (Smith et al., 2012) in MRTrix to obtain whole-brain probabilistic streamlines. IFOD2 is an established tractography method with good performance in constructing crossing fibers (Tournier et al., 2010). ACT has been shown to prevent some of the known false positives in tractography. Compared with our previous results, the use of ACT led to some changes in the structural connectivity, and we have updated our statistical results throughout the manuscript. Nevertheless, our main results (e.g., the relationships between SC-ISV and R1, and the PCA analysis on connectome microstructure) remain unchanged.

We combined several additional anatomical constraints to minimize false positives in streamlines tractography (Schirner et al., 2015). First, all tracks were initiated from a seed mask that includes grey-matter/white-matter boundary voxels of the region from Freesurfer segmentation. Second, we used the white-matter volume as an inclusion mask. Streamlines that leaves the white-matter mask (e.g., entering grey-matter or ventricles) were terminated and rejected. Third, we used a clustering algorithm to remove individual streamlines that can be considered as anatomical outliers (e.g., Côté et al., 2015).

4. How does the interindividual variability compare to intra-individual variability? Is it possible that the variability between people is comparable to the variability of the same brain scanned at different time points?

The Reviewer raised an important issue, and we have addressed it in the Revision.

First, we agree with the Reviewer that intra-subject variability can affect the calculation of inter-subject variability of functional connectivity (Mueller et al., 2013). In all our MEG analysis, for each participant, intra-individual variability between two scanning sessions acquired on different days were regressed out. Our original manuscript analysed fMRI data from a single session. Nevertheless, we did not acquire multiple fMRI resting-state scans from those participants to correct for fMRI intra-individual variability. To ensure that our results are not confounded by potential intra-individual variability, we have removed all fMRI analyses in the revised manuscript and focus on MEG- and DWI-based ISV. We think the revised manuscript highlighted our new contributions on MEG- and SC-ISV, considering that the ISV of fMRI connectivity has now been reported in several papers.

Second, although functional connectivity contains rich temporal-dependent fluctuations (Hutchison et al., 2013) and variabilities between sessions, DWI-based structural connectivity exhibits good test-retest reliability between sessions. Structural connectivity based on probabilistic tractography (i.e., the method used in the current study) has superior reliability than deterministic approaches (Bonilha et al., 2015). In the Revision, we further demonstrated that the ISV of structural connectivity is highly

consistent between two different cohorts using different MR systems and DWI sequences: our own dataset and the Cam-CAN dataset collected at Cambridge (see Point 1 to Reviewer 1).

In sum, the revised manuscript reports (1) the ISV of structural connectivity that is replicable between independent cohorts using different scanners and sequences, and (2) the ISV of MEG functional connectivity that is corrected for intra-individual variability.

5. How do the results of structural and functional connectomes showing lower variability in sensory and visual cortices sit in comparison to the previous literature that showed high variability in these regions? Some references that come to mind [...]:

We thank the Reviewer for this valuable comment and relevant references, which we have discussed in the Revision (line 659).

“ [...] Brain variability exhibits in different forms (Gordon and Nelson, 2021). For example, one can quantify anatomical variability as the amount of deformation between individual brains and a group template. In both humans and non-human primates, the anatomical variability in visual and frontal areas was higher than in other brain regions (Croxson et al., 2018). fMRI localisation and cytoarchitectonic classification studies showed that visual (Uylings et al., 2005) and motor cortices (Yousry, 1997; Geyer et al., 1999) have high morphometric variability. These brain variability measures differ from the results of the current study, which focused on the ISV of structural and functional connectivity strength. [...] Similar spatial patterns of the variability of connectivity strength have been reported elsewhere using different imaging modality, acquisition parameters, atlases and preprocessing methods (Mueller et al., 2013; Chamberland et al., 2017; Stoecklein et al., 2020; Mansour L et al., 2021). ”

6. Finally, in the absence of any cognitive and clinical data, how can the authors be sure these results are meaningful beyond the immediate study?

Our new ISV results from of the Cam-CAN data indicates that our analyses are applicable, and more importantly, generalizable to other cohorts. In Discussion, we highlighted that an important next step is to link the variability of brain connectivity to that of demographical and neurological/neuropsychiatric variables (line 816).

Furthermore, we have included new results to probe the functional significance of the ISV of brain connectivity (lines 451-458 and 606-620). In this new analysis (see the figure below, which is the new Figure 6 in the manuscript), we compared the SC-ISV between multiple-demand (MD) regions and the rest of the brain. The MD network includes brain regions with integrative properties in response to cognitively demanding tasks (Duncan, 2010). We used a recent definition of the extended MD network on the HCP-MMP atlas (Assem et al., 2020b), which includes 10 regions per hemisphere as the core MD network and 17 additional regions as the penumbra. In both Cohort 1 and Cam-CAN datasets, the ISV of structural connectivity is higher in the core MD regions than the rest of the cortex. Therefore, cortical regions that are essential for cognitive flexibility and integration not only have stronger interconnectivity (Assem et al., 2020b), but also exhibit greater inter-subject variability.

We have not conducted this analysis on MEG-ISV, because the down-sampled atlas that optimized for MEG source-level signal-to-noise ratio (Tait et al., 2021) merges some ROIs belongs to MD and non-MD categories.

Figure. SC-ISV in the multiple demand (MD) network. **A.** The extended MD network includes the core MD regions and the penumbra. All MD regions were defined in the HCP-MMP atlas

(Assem et al., 2020b). The mean SC-ISV of the core MD regions, the penumbra and non-MD regions from Cohort 1 (left) and Cam-CAN (right) datasets. Errorbars denotes 95% confidence interval. Asterisks denote statistical significance ($p < 0.05$) from a two-sided permutation test (5,000 permutations). This figure is included in the Revision (the new Figure 6).

Reviewer 2

1. The study certainly addresses an interesting question about cross-modal correspondence depends on inter-individual variability. The data processing and most of the analyses are carried out at a high technical standard. At the same time, there a number of conceptual and methodological points that should be addressed.

We thank the Reviewer for their encouraging comment. In the Revision, we have clarified our main contributions in Discussion (see Point 2 below). We have also addressed methodological issues listed below.

2. It is not clear what the overall conclusion is. What do the authors make of the differing patterns of ISVs? Why should areas with greater R1 have lower ISVs? There is a notable lack of conceptual coherence.

We have made substantial updates throughout the manuscript to improve its conceptual coherence. With the addition of new participants and a new independent DWI dataset from the Cam-CAN cohort (<https://www.cam-can.org>) (see sections 2.1 and 2.11, also see Point 1 to Reviewer 1), we have re-structured all figures to highlight our main contributions.

- Figure 1 – schematics of the method
- Figure 2 – the spatial heterogeneity of SC-ISV and MEG-ISV
- Figure 3 – cross-modal alignments between SC-ISV and MEG-ISV
- Figure 4 – The relations between grey-matter microstructure (i.e., R1) and ISV
- Figure 5 – The relations between white-matter microstructure and MEG-ISV
- Figure 6 – The SC-ISV in the multiple-demand network.

Among those results, Figures 2 and 3 characterised the ISV. Figures 4 and 5 examined the link between ISV and brain microstructural measures. Figure 6 considered potential functional significance of ISV.

We have removed fMRI results from the manuscript, because we cannot rule out that our previous fMRI results were not affected by within-subject variability (see Point 4

to Reviewer 1). Although our DWI data are also from a single session, we are confident that our results are robust, as we have replicated them in the Cam-CAN dataset (including the spatial heterogeneity of SC-ISV and its correlation to the R1 map). We feel that the revised manuscript highlights better the main contributions of the current study.

Furthermore, we have revised the Discussion to address the issues raised by the Reviewer. We listed the main changes below together with the line numbers in the manuscript.

Overall conclusion

(line 648) DWI and MEG connectivity variability is consistent in selective cortical clusters, as supported by significant cross-modal ISV alignments. Across the cortex, The ISV of structural and beta-band MEG connectivity is negatively associated with cortical myelin content. For beta- and gamma-band MEG connectivity, its ISV further relates to tissue microstructure estimated from white-matter compartmental models. Our findings extended the current understanding of brain connectivity variability in multiple modalities and suggested the important roles of tissue microstructure in shaping connectivity variability.

What do we make of the different patterns of ISVs, and why should/does R1 correlate with ISV?

SC-ISV (lines 659-691)

The SC-ISV in heteromodal association cortices (frontal, parietal and cingulate areas) are higher than those in unimodal sensorimotor cortices (visual, auditory, somatosensory and motor areas). We further replicated this result in an independent dataset (Cam-CAN). [...]

The spatial distribution of SC-ISV carries functional significance: the SC-ISV is high in the core MD regions. Core MD regions in the frontoparietal cortex are essential for cognitive flexibility and integration, and they have been linked to the individual difference in memory and fluid intelligence (Woolgar et al., 2018; Assem et al., 2020a). Here, we extended previous findings that core MD regions have strong

interconnectivity (Assem et al., 2020b), by showing core MD regions to have greater connectivity variability than the rest of the cortex.

The anatomical signature of this unimodal-heteromodal distinction further resembles the principal gradient of cortical hierarchy that is related to synaptic physiology and cytoarchitecture (Huntenburg et al., 2017; Burt et al., 2018). [...]

MEG-ISV (lines 700-736)

The ISV of MEG functional connectivity had similar spatial distributions in theta, alpha and beta band oscillatory connectivity, in that frontal connections have higher ISV. Between the three lower frequency bands, the relative strength of the MEG-ISV in the visual, motor and temporal cortices differed. [...] Gamma-band MEG-ISV has a distinct spatial signature, possibly owing to its different connectivity pattern within and between large resting state networks (Samogin et al., 2020). By definition, regions with predominately reliable connections would lead to smaller ISV than those with weak and variable connections. Hence, the frequency-dependent MEG-ISV patterns can also be related to different neural oscillators underpinning MEG functional connectivity at rest, which have been shown to reliably represent intrinsic fluctuations between spatially distant brain regions (Brookes et al., 2011). [...]

We further observed the alignment of ISVs between modalities. That is, the extent of dissimilarity between different individuals is preserved between structural and functional connectivity in a cluster-specific manner. This cross-modal alignment of ISV was significant across all MEG frequency bands in the lateral temporal cortex. For beta-band MEG-ISV, a large group of frontoparietal clusters showed significant ISV alignment to SC-ISV. The ISV alignment could be due to common neuroanatomical or neurobiological constraints modulating both structural and functional connectivity, including major white-matter tracts, cortical folding, sulcal depth (Hill et al., 2010a; Mueller et al., 2013) and late maturation (Hill et al., 2010b). For example, the human superior longitudinal fasciculus (SLF) connects frontal to parietal areas and modulates control networks (de Schotten et al., 2011), and the individual difference of the lateral SLF is associated with the long-range synchronization of beta-band oscillation during visuo-spatial attention processes (Quentin et al., 2015), supporting the potential alignment between structural and beta-band oscillatory variability.

ISV and R1

[...] Both Cohort 1 and Cam-CAN data showed that SC-ISV negatively correlate with the R1 map (after correction for spatial autocorrelation, see Point 9 to Reviewer 2). For MEG, the ISV in the beta-band showed a similar negative correlation. Quantitative R1 maps provide a good proxy of cortical myelin content (Marques et al., 2017). This leads to two potential interpretations of the robust R1-ISV associations. (1) Both ISV and R1 maps follow the principal gradient of cortical hierarchy, and it is an emergent property of large-scale topography of the human brain. (2) Intracortical myelination does directly impact on the variability of connectivity, and hence the spatial heterogeneity of ISV accompanies the regional difference of intracortical myelination, which is sensitive to *in vivo* MR contrasts such as R1 and T1w/T2w ratio.

Although these two propositions are not mutually exclusive, there are evidence supporting a direct impact of intracortical myelination on connectivity variability. Myelin related factors reduce synaptic plasticity by inhibiting neurite growth such as new axonal growth and synapse formation (McGee, 2005). Lightly myelinated frontal and parietal regions require higher aerobic glycolysis than lightly myelinated areas (Glasser et al., 2014). The combination of low myelin content and high aerobic glycolysis may be a characteristic neurobiological feature of the association cortex, enabling adaptable and plastic neural circuitry, which in turn leads to high ISV in functional and structural connectivity. However, one would need a direct experimental manipulation, likely with animal models, to confirm this hypothesis.

3. Can you just merge Figures 2-4? Also label colormaps for panels a, and provide more info for panels b.

Following the Reviewer's suggestion, we have merged all results of DWI and MEG ISVs in the new Figure 2. We have added the label for the colormap of connectivity matrices (i.e., connection probability for structural connectivity and Person correlation coefficient for MEG functional connectivity). In the figure caption, we provided further information for the brain maps.

4. Throughout the manuscript, the parcellation does not look like MMP – it looks like it has substantially fewer regions. Can the authors confirm?

In our original manuscript, we plotted the ISV values at the cortical cluster level. In the Revision, we have updated all brain maps (Figures 2-5, see also the Figure below in point 6 to Reviewer 2) to render the values of individual ROIs.

5. I don't think that it is sufficient to just show alpha and beta based on one prior paper. Please show results for all the canonical frequency bands, from delta to gamma.

In line with the Reviewer's comment, we have reported results from all four MEG frequency bands from theta to gamma (Figures 2-5), including the analyses in relation to R1 values and microstructural metrics. The figure below shows the ISV of MEG functional connectivity (which is a part of the new Figure 2 in the manuscript). Theta-band MEG-ISV does not relate to any microstructural measures. Hence, we have not conducted analyses on the slower delta-band MEG connectivity, as we expect it would yield similar results as the theta-band MEG. Keeping the four frequency bands also simplify the results and figures. We are happy to conduct additional analyses on the delta-band MEG data if the Reviewer considers them essential.

Figure. MEG-ISV in theta (4-8 Hz), alpha (8-13 Hz), beta (13-30 Hz) and gamma (30-100 Hz) bands. The brain maps (upper left) rendered the z-score of each ROI's ISV on the cortical surface. The barplot (right) showed the mean ISV of each of the 22 cortical clusters, which was estimated from all ROIs within each cluster. Error bars denote 95% confidence intervals. The matrix (lower left) denoted the group average of connectomes.

6. The labels in the bars have acronyms that are a bit mysterious. For examples, what is FRN?

We have removed the acronym FRN. In the revision, we used the common acronyms of the 22 cortical clusters that are defined in the HCP atlas. The figure below (included as Supplementary Figure 3) lists the full name and the acronym of each cortical cluster defined in the original HCP-MMP atlas (Glasser et al., 2016), as well as each cluster's anatomical location.

Figure. 22 cortical clusters defined in the HCP-MMP atlas (Glasser et al., 2016). Clusters close to each other are rendered in similar colors. The full name given in (Glasser et al., 2016) and the acronym of each cluster were shown. For the cluster of *MT+ and neighbouring visual areas*, we used the acronym MT+/LOC as the neighbouring areas mainly include the lateral occipital cortex (LOC). For the cluster of *posterior opercular cortex*, we used the acronym S2, as the second somatosensory area makes most of this cluster.

7. When comparing ISVs, why not correlate the ISV maps instead?

We have clarified this issue in the Revision (line 415). Previous studies showed no strong relationship between the ISVs of DWI-based structural connectivity and fMRI-based functional connectivity (Chamberland et al., 2017). The current study tested a different hypothesis. That is, for a given cortical cluster, do a pair of participants with higher structural connectivity difference (quantified by the cosine distance) also exhibit higher MEG functional connectivity difference? This analysis allows us to examine the alignment of interindividual difference of brain connectivity at cortical cluster level.

8. Combine Figs 5 and 6. Also missing caption for Figure 6.

We have corrected these issues. The new Figure 3 in the Revision reported all across-modal alignments between MEG-ISV and SC-ISV.

9. Figure 7: please use spatial autocorrelation -preserving null models to assess the significance of the correlation.

We thank the Reviewer for this important suggestion. In the Revision, for each ROI-based correlational analysis, we used *brainsmash* (Burt et al., 2020) to generate 5,000 surrogate brain maps with the same spatial autocorrelation as in the observed data. The surrogate maps were then used to obtain corrected p -values for all tests of spatial correspondence between brain maps.

We have outlined the method and procedure of generating surrogate brain maps in the Revision (line 439). Throughout the manuscript, we have reported both the uncorrected and corrected (with the notion “SA-corrected”) p -values. The figure below showed an example of the null distributions with and without preserving spatial autocorrelation in the tests of the correlation between R1 and SC-ISV maps (i.e., Figure 4B in the manuscript). For completeness, we have reported null distributions of all tests in the new Supplementary Figures S6 and S7.

As reported elsewhere (Markello and Misic, 2021), correction for spatial autocorrelation is critical in evaluating correlations between brain maps, because the non-independency among ROIs inflates p -values and increases type I errors. In the Revision, we have therefore removed the original mediation analysis (between R1, MEG-ISV and microstructural metrics), because we have not found an established method for correcting spatial autocorrelation in mediation analysis.

Figure. Null distributions and observed Spearman correlation coefficients between the R1 cortical map from Cohort 2 7T data and SC-ISV maps (*left*: Cohort 1 data; *right*: Cam-CAN data). SA-preserving null distributions were obtained from correlations with 5,000 surrogate maps with the same spatial autocorrelation as in the empirical brain maps. SA-independent null distributions were obtained from correlations with 5,000 iterations of randomly shuffled values. Dashed vertical line indicates the empirical Spearman's correlation coefficient.

Reference

- Assaf Y, Basser PJ (2005) Composite hindered and restricted model of diffusion (CHARMED) MR imaging of the human brain. *Neuroimage* 27:48–58.
- Assem M, Blank IA, Mineroff Z, Ademoğlu A, Fedorenko E (2020a) Activity in the fronto-parietal multiple-demand network is robustly associated with individual differences in working memory and fluid intelligence. *Cortex* 131:1–16.
- Assem M, Glasser MF, Van Essen DC, Duncan J (2020b) A Domain-General Cognitive Core Defined in Multimodally Parcellated Human Cortex. *Cereb Cortex* 30:4361–4380.
- Bonilha L, Gleichgerrcht E, Fridriksson J, Rorden C, Breedlove JL, Nesland T, Paulus W, Helms G, Focke NK (2015) Reproducibility of the Structural Brain Connectome Derived from Diffusion Tensor Imaging Hayasaka S, ed. *PLoS One* 10:e0135247.
- Brookes MJ, Hale JR, Zumer JM, Stevenson CM, Francis ST, Barnes GR, Owen JP, Morris PG, Nagarajan SS (2011) Measuring functional connectivity using MEG: Methodology and comparison with fcMRI. *Neuroimage* 56:1082–1104.
- Burt JB, Demirtaş M, Eckner WJ, Navejar NM, Ji JL, Martin WJ, Bernacchia A, Anticevic A, Murray JD (2018) Hierarchy of transcriptomic specialization across human cortex captured by structural neuroimaging topography. *Nat Neurosci* 21:1251–1259.
- Burt JB, Helmer M, Shinn M, Anticevic A, Murray JD (2020) Generative modeling of brain maps with spatial autocorrelation. *Neuroimage* 220:117038.
- Chamberland M, Girard G, Bernier M, Fortin D, Descoteaux M, Whittingstall K (2017) On the Origin of Individual Functional Connectivity Variability: The Role of White Matter Architecture. *Brain Connect* 7:491–503.
- Chamberland M, Raven EP, Gene S, Duffy K, Descoteaux M, Parker GD, Tax CMW, Jones DK (2019) Dimensionality reduction of diffusion MRI measures for improved tractometry of the human brain. *Neuroimage* 200:89–100.

- Côté MA, Garyfallidis E, Laroche H, Descoteaux M (2015) Cleaning up the mess: tractography outlier removal using hierarchical QuickBundles clustering. In: Proceedings of: International Society of Magnetic Resonance in Medicine, pp 2844.
- Crosson PL, Forkel SJ, Cerliani L, Thiebaut de Schotten M (2018) Structural Variability Across the Primate Brain: A Cross-Species Comparison. *Cereb Cortex* 28:3829–3841.
- de Schotten MT, Dell’Acqua F, Forkel SJ, Simmons A, Vergani F, Murphy DGM, Catani M (2011) A lateralized brain network for visuospatial attention. *Nat Neurosci* 14:1245–1246.
- Duncan J (2010) The multiple-demand (MD) system of the primate brain: mental programs for intelligent behaviour. *Trends Cogn Sci* 14:172–179.
- Eggenschwiler F, Kober T, Magill AW, Gruetter R, Marques JP (2012) SA2RAGE: A new sequence for fast B1+-mapping. *Magn Reson Med* 67:1609–1619.
- Fan Q et al. (2022) Mapping the human connectome using diffusion MRI at 300 mT/m gradient strength: Methodological advances and scientific impact. *Neuroimage* 254:118958.
- Geyer S, Schleicher A, Zilles K (1999) Areas 3a, 3b, and 1 of Human Primary Somatosensory Cortex. *Neuroimage* 10:63–83.
- Glasser MF, Coalson TS, Robinson EC, Hacker CD, Harwell J, Yacoub E, Ugurbil K, Andersson J, Beckmann CF, Jenkinson M, Smith SM, Van Essen DC (2016) A multi-modal parcellation of human cerebral cortex. *Nature* 536:171–178.
- Glasser MF, Goyal MS, Preuss TM, Raichle ME, Van Essen DC (2014) Trends and properties of human cerebral cortex: Correlations with cortical myelin content. *Neuroimage* 93:165–175.
- Gordon EM, Nelson SM (2021) Three types of individual variation in brain networks revealed by single-subject functional connectivity analyses. *Curr Opin Behav Sci* 40:79–86.

- Hill J, Dierker D, Neil J, Inder T, Knutsen A, Harwell J, Coalson T, Van Essen D (2010a) A Surface-Based Analysis of Hemispheric Asymmetries and Folding of Cerebral Cortex in Term-Born Human Infants. *J Neurosci* 30:2268–2276.
- Hill J, Inder T, Neil J, Dierker D, Harwell J, Van Essen D (2010b) Similar patterns of cortical expansion during human development and evolution. *Proc Natl Acad Sci* 107:13135–13140.
- Huntenburg JM, Bazin P-L, Goulas A, Tardif CL, Villringer A, Margulies DS (2017) A Systematic Relationship Between Functional Connectivity and Intracortical Myelin in the Human Cerebral Cortex. *Cereb Cortex* 27:981–997.
- Hutchison RM, Womelsdorf T, Allen EA, Bandettini PA, Calhoun VD, Corbetta M, Della Penna S, Duyn JH, Glover GH, Gonzalez-Castillo J, Handwerker DA, Keilholz S, Kiviniemi V, Leopold DA, de Pasquale F, Sporns O, Walter M, Chang C (2013) Dynamic functional connectivity: Promise, issues, and interpretations. *Neuroimage* 80:360–378.
- Jones DK, Alexander DC, Bowtell R, Cercignani M, Dell’Acqua F, McHugh DJ, Miller KL, Palombo M, Parker GJM, Rudrapatna US, Tax CMW (2018) Microstructural imaging of the human brain with a ‘super-scanner’: 10 key advantages of ultra-strong gradients for diffusion MRI. *Neuroimage* 182:8–38.
- Maffei C, Sarubbo S, Jovicich J (2019) Diffusion-based tractography atlas of the human acoustic radiation. *Sci Rep* 9:4046.
- Mansour L S, Tian Y, Yeo BTT, Cropley V, Zalesky A (2021) High-resolution connectomic fingerprints: Mapping neural identity and behavior. *Neuroimage* 229:117695.
- Markello RD, Misic B (2021) Comparing spatial null models for brain maps. *Neuroimage* 236:118052.
- Marques JP, Gruetter R (2013) New Developments and Applications of the MP2RAGE Sequence - Focusing the Contrast and High Spatial Resolution R1 Mapping. *PLoS One* 8.

- Marques JP, Khabipova D, Gruetter R (2017) Studying cyto and myeloarchitecture of the human cortex at ultra-high field with quantitative imaging: R_1 , R_2^* and magnetic susceptibility. *Neuroimage* 147:152–163.
- Marques JP, Kober T, Krueger G, van der Zwaag W, Van de Moortele PF, Gruetter R (2010) MP2RAGE, a self bias-field corrected sequence for improved segmentation and T1-mapping at high field. *Neuroimage* 49:1271–1281.
- McGee AW (2005) Experience-Driven Plasticity of Visual Cortex Limited by Myelin and Nogo Receptor. *Science* (80-) 309:2222–2226.
- Mueller S, Wang D, Fox MD, Yeo BTT, Sepulcre J, Sabuncu MR, Shafee R, Lu J, Liu H (2013) Individual Variability in Functional Connectivity Architecture of the Human Brain. *Neuron* 77:586–595.
- O'Brien KR, Kober T, Hagmann P, Maeder P, Marques J, Lazeyras F, Krueger G, Roche A (2014) Robust T1-Weighted Structural Brain Imaging and Morphometry at 7T Using MP2RAGE Margulies D, ed. *PLoS One* 9:e99676.
- Quentin R, Chanes L, Vernet M, Valero-Cabré A (2015) Fronto-Parietal Anatomical Connections Influence the Modulation of Conscious Visual Perception by High-Beta Frontal Oscillatory Activity. *Cereb Cortex* 25:2095–2101.
- Samogin J, Marino M, Porcaro C, Wenderoth N, Dupont P, Swinnen SP, Mantini D (2020) Frequency-dependent functional connectivity in resting state networks. *Hum Brain Mapp* 41:5187–5198.
- Schirner M, Rothmeier S, Jirsa VK, McIntosh AR, Ritter P (2015) An automated pipeline for constructing personalized virtual brains from multimodal neuroimaging data. *Neuroimage* 117:343–357.
- Schmierer K, Scaravilli F, Altmann DR, Barker GJ, Miller DH (2004) Magnetization transfer ratio and myelin in postmortem multiple sclerosis brain. *Ann Neurol* 56:407–415.
- Schmierer K, Wheeler-Kingshott CAM, Tozer DJ, Boulby PA, Parkes HG, Yousry TA, Scaravilli F, Barker GJ, Tofts PS, Miller DH (2008) Quantitative magnetic

resonance of postmortem multiple sclerosis brain before and after fixation. *Magn Reson Med* 59:268–277.

Smith RE, Tournier J-D, Calamante F, Connelly A (2012) Anatomically-constrained tractography: Improved diffusion MRI streamlines tractography through effective use of anatomical information. *Neuroimage* 62:1924–1938.

Sotiropoulos SN, Jbabdi S, Xu J, Andersson JL, Moeller S, Auerbach EJ, Glasser MF, Hernandez M, Sapiro G, Jenkinson M, Feinberg DA, Yacoub E, Lenglet C, Van Essen DC, Ugurbil K, Behrens TEJ, WU-Minn HCP Consortium (2013) Advances in diffusion MRI acquisition and processing in the Human Connectome Project. *Neuroimage* 80:125–143.

Stoecklein S, Hilgendorff A, Li M, Förster K, Flemmer AW, Galiè F, Wunderlich S, Wang D, Stein S, Ehrhardt H, Dietrich O, Zou Q, Zhou S, Ertl-Wagner B, Liu H (2020) Variable functional connectivity architecture of the preterm human brain: Impact of developmental cortical expansion and maturation. *Proc Natl Acad Sci* 117:1201–1206.

Tait L, Özkan A, Szul MJ, Zhang J (2021) A systematic evaluation of source reconstruction of resting MEG of the human brain with a new high-resolution atlas: Performance, precision, and parcellation. *Hum Brain Mapp* 42:4685–4707.

Taylor JR, Williams N, Cusack R, Auer T, Shafto MA, Dixon M, Tyler LK, Cam-CAN, Henson RN (2017) The Cambridge Centre for Ageing and Neuroscience (Cam-CAN) data repository: Structural and functional MRI, MEG, and cognitive data from a cross-sectional adult lifespan sample. *Neuroimage* 144:262–269.

Tournier JD, Calamante F, Connelly A (2010) Improved probabilistic streamlines tractography by 2nd order integration over fibre orientation distributions. In: *Proceedings of the International Society for Magnetic Resonance in Medicine*, pp 1670.

Uylings HBM, Rajkowska G, Sanz-Arigo E, Amunts K, Zilles K (2005) Consequences of large interindividual variability for human brain atlases: converging macroscopical imaging and microscopical neuroanatomy. *Anat Embryol (Berl)*

210:423–431.

Woolgar A, Duncan J, Manes F, Fedorenko E (2018) Fluid intelligence is supported by the multiple-demand system not the language system. *Nat Hum Behav* 2:200–204.

Xu T, Sturgeon D, Ramirez JSB, Froudish-Walsh S, Margulies DS, Schroeder CE, Fair DA, Milham MP (2019) Interindividual Variability of Functional Connectivity in Awake and Anesthetized Rhesus Macaque Monkeys. *Biol Psychiatry Cogn Neurosci Neuroimaging* 4:543–553.

Yousry T (1997) Localization of the motor hand area to a knob on the precentral gyrus. A new landmark. *Brain* 120:141–157.

Zhang H, Schneider T, Wheeler-Kingshott CA, Alexander DC (2012) NODDI: Practical in vivo neurite orientation dispersion and density imaging of the human brain. *Neuroimage* 61:1000–1016.

REVIEWERS' COMMENTS:

Reviewer #1 (Remarks to the Author):

The authors have considerably revised the manuscript and addressed the comments raised sufficiently.

Reviewer #2 (Remarks to the Author):

The authors have comprehensively addressed all my suggestions and I recommend publication.